# Wet and dry spells in Senegal: Comparison of detection based on satellite products, reanalysis and in-situ estimates

Cheikh Modou Noreyni Fall[1], Christophe Lavaysse[2], Mamadou Simina Drame[1], Geremy Panthou[2], and Amadou Thierno Gaye[1]

[1]Laboratoire de Physique de l'Atmosphère et de l'Océan—Simeon Fongang, ESP, Univ. Cheikh Anta Diop, Dakar, Senegal
[2]Institut des Géosciences de l'Environnement IGE, Univ. Grenoble Alpes, IRD, CNRS, Grenoble INP, 38000 Grenoble, France

**Correspondence:** Noreyni Fall (noreyni27@gmail.com)

**Abstract.** In this study, the detection and characteristics of dry/wet spells (defined as episodes when precipitation is abnormally low or high compared to usual climatology) drawn from several datasets are compared for Senegal. Here four datasets are based on satellite data (TRMM-3B42 V7, CMORPH V1.0, TAMSAT V3, CHIRPS V2. 0), two on reanalysis products (NCEP-CFSR, ERA5) and three on raingauge observations: CPC Unified V1.0/RT and a 65 raingauge network regridded by using two kriging methods, namely ordinary kriging (OK) and block kriging (BK). All datasets were converted to the same spatiotemporal resolution: daily cumulated rainfall on a regular 0.25 degree grid. The BK dataset was used as a reference. Despite strong agreement between the datasets on the spatial variability of cumulative seasonal rainfall (correlations ranging from 0.94 to 0.99), there were significant disparities in dry/wet spell. The occurrence of dry spells is less in products using infrared measurement techniques than in products coupling infrared and microwave, pointing to more frequent dry spell events. All datasets show that dry spells appear to be more frequent at the start and end of rainy seasons. Thus, dry spell occurrences have a major influence on the duration of the rainy season, in particular through 'false onset' or 'early cessation' of seasons. The amplitude of wet spells shows greatest variation between datasets. Indeed, these major wet spells appear more intense in the OK and TRMM datasets than in the others. Lastly, the products indicate a similar wet spell frequency occurring at the height of the West African monsoon. Our findings provide guidance in choosing the most suitable datasets for implementing early warning systems (EWS) using a multi-risk approach and integrating effective dry/wet spell indicators for monitoring and detection of extreme events.

Several studies on climate change predict intensification of hydrological cycles, and thus increased probability of heavy rainfall and dry periods due to global warming (Held and Soden, 2008; Giorgi et al., 2011; Trenberth, 2011; Kendon et al., 2019; Berthou et al., 2019). An increase in extreme events is a major phenomenon accompanying Sahel rainfall recovery (Alhassane et al., 2013; Descroix et al., 2016; Panthou et al., 2014, 2018; Taylor et al., 2017; Wilcox et al., 2018). An estimated 1.7 million people have been affected by floods in Benin, Burkina Faso, Chad, Ghana, Niger, Nigeria and Togo since the second

half of the years 2000 (Sarr, 2012). In 2009, Benin, Burkina Faso, Niger and Senegal all reported major flooding (Engel et al., 2017; Fowe et al., 2018; Salack et al., 2018), while heavy rains impacted more than 80% of Nigeria in 2012. Extreme events occurred as well in Burkina Faso, including record rainfall of 263 mm in Ouagadougou in September 2009 (Lafore et al., 2017). In Senegal, over 26 people died from direct or indirect repercussions of an extreme rainfall event on August 26, 2012,

with 161 mm recorded in less than 3 hours (Sagna et al., 2015; Young et al., 2019). On the other hand, UN agencies judged that over 16 million people in Mali, Sudan, Niger, Burkina Faso, Senegal and Chad were affected by the 2012 drought (UCDP, 2017). In 2014, severe drought hit several areas of Senegal, leading to 16.5 USD million in funding from the African Risk Capacity (ARC, 2014) for the Senegalese government. In 2018, according to the World Food Programme (WFP), Senegal was one of seven Sahelian countries with significant increase in numbers of food-insecure people, from 314,600 to 548,000 in the

2018 lean season (WFP, 2018). In such a context of high risk combined with extreme hydro-meteorological events and highly vulnerable populations, a better understanding of multi-scale rainfall regime variability is essential (Le Barbé et al., 2002; Lebel and Ali, 2009; Nicholson, 2013; Dione et al., 2014; Yeni and Alpas, 2017).

Several authors using raingauges, rainfall estimates from satellite imagery and Numerical Weather Prediction (NWP) focused on the multi-scale variability of these potentially high-impact events (Washington et al., 2006; Sane et al., 2018; Nicholson

et al., 2018). Indeed, with partial rainfall recovery, the Sahel experiences mixed dry/wet seasonal rainfall features known as hybrid rainy seasons (Salack et al., 2016). These hybrid rainy seasons illustrating 'Hydroclimatic Intensity' are what Giorgi et al. (2011) defined as a more extreme hydrological climate, with longer dry spells and more intense rainfall (Trenberth et al., 2003; Trenberth, 2011). However, a deeper analysis of the components of these hybrid seasons is still lacking, even though other studies analyzed the spatial/temporal variability of dry/wet spells over West Africa and showed a close correlation to

West African monsoon spatial/temporal variability (Froidurot and Diedhiou, 2017). Furthermore, in terms of seasonal cycles, these longer dry spells generally occur at the start and end of rainy seasons, making them crucial to agro-climatic monitoring (Salack et al., 2013). The study by Dieng et al. (2008) showed that an earlier (later) long dry spell is associated with higher (lower) cumulative seasonal rainfall in July through September in northern Senegal. However, this correlation is less distinct in the south of the country. Understanding and monitoring such high-impact events can yield important applications for agronomy

and disaster risk mitigation.

Very few studies have compared the performances of satellite imagery, reanalysis products and ground observations in detection of the distribution of dry/wet spells over the Sahel. The scant comparative studies conducted in Africa have focused mainly on interannual variability of seasonal rainfall amounts (Thorne et al., 2001; Ali et al., 2005). Tropical Applications of Meteorology Using Satellite Data and Ground-Based Observations (TAMSAT) has proven successful in many areas of Africa

despite its relatively simple algorithm (Thorne et al., 2001; Jobart et al., 2011; Dinku et al., 2007; Maidment et al., 2013). The CPC MORPHing technique (CMORPH) appears to confirm gauge data in Ethiopia but greatly underestimate rainfall amounts in the Sahel (Jobard et al., 2010). Meanwhile, past studies have also demonstrated that Tropical Rainfall Measuring Mission (TRMM 3B42) data adequately captures spatial variations in annual and seasonal precipitations (Xu et al., 2019). Nevertheless, it tends to overestimate trace precipitation and underestimate torrential precipitation on daily scales owing to

inadequate detection capability (Xu et al., 2019; Shuhong et al., 2019). Often reanalyses are of lesser quality in the tropics,

particularly in Africa where there are unfortunately few in situ observations. However, this appears to have improved since satellite observations have been more widely used, since they are incorporated into assimilation systems (Parker, 2016).

Thus, there is a great need for a broad inter-comparative study of these products in a region that is not only well documented but also equipped with a high density raingauge network. This paper aims to set up an inter-comparison between several products resulting from observations, satellite data and models. More specifically, the idea is to compare their ability to detect potentially high-impact dry/wet events in Senegal. The reference dataset chosen for this study is block kriging (BK), despite the imprecision of the network of surface observations and uncertainties related to kriging techniques. Although OK produces the best point-based estimates, in cases where nugget variance is great, interpolated surfaces may be subject to local discontinuities, consequently troubling longer range spatial variations. BK circumvents this by computing averaged estimates over areas or volumes (albeit at the cost of reduced spatial resolution). BK estimates may also be more realistic since data from one point usually represents the area around it. In this paper, potentially high impact indicators are defined and characterized. Here we use the term "potentially high impact indicators" to illustrate the extreme dry/wet spells subject to this analysis. This term is used to better encompass vulnerability, exposure of populations and risks of hazard. Our paper is structured in sections, as follows. Section 1 describes the data and methodology used in our analysis, Section 2 presents the main findings and results of statistical tests, and Sections 3 and 4 concludes the paper with a discussion of our main findings and wider implications.

## 1 Data and Methodological Approach

### 1.1 Rain Gauge Data and Kriging Methods

Daily rainfall data were provided by the National Meteorological Service of Senegal (ANACIM) for 65 locations covering the period 1991-2010 (Fig. 1). Two levels of quality control were carried out for an objective verification of homogeneity. One manual check of dubious records was done, followed by other checks including verification of station locations, identification of redundant data, identification of outliers, tests comparing neighboring stations and examination of suspicious zero values (i.e. missing data or no precipitation). The 1991-2010 time span is the longest period having the maximum number of reliable stations with sufficient spatial coverage allowing study objectives to be met. Even so, the geographical distribution of this network shows a strong east-west imbalance. An overview of the network shows more raingauges in the peanut-growing basin (central western zone) than elsewhere in the country (See Fig 1). This is due to the intensive agricultural production in this area where rainfall imposes a limit on economic activity. Because it is difficult to compare raingauge (point measurement) data with satellite datasets, raingauge data was gridded into a $0.25 \times 0.25$ degree resolution using two different kriging methods: ordinary kriging (OK) and block kriging (BK). Several studies have shown these kriging techniques to be among the most efficient interpolation methods (Creutin and Obled, 1982; Tabios and Salas, 1985; Goovaerts, 2000). Because different techniques do exist and some inherent uncertainties remain, two kriging methods were used in this study. OK was used to estimate a value at

a point in a region for which there is a known variogram, applying data near the estimation location (Myers, 1997; Chen et al., 2008; Wei et al., 2009). Equation 1 gives a value for rain estimated by ordinary kriging.

$$Z^k = \sum_{i=1}^{n} \lambda_i Z_i^o \tag{1}$$

where $Z^k$ and $Z^o$ respectively represent the rainfall estimate and the observed raingauge values, lambda is weightings assigned to $n$ available observations. The $\lambda_i$ kriging weightings are obtained by configuring an optimization scheme containing $n+1$ simultaneous linear equations. These equations are derived from the standard variogram models for the distance separating sampling points from target locations using the Lagrange multiplier. The second kriging technique, BK, uses a moving district or a block of given dimensions to estimate the average $Z$ value over a surface (Lloyd and Atkinson, 2001; Maidment et al., 2013). The average value of $Z$ attribute over a $V$ block centered at the block's mean value is computed using equation 2.

$$Z_v^k = \frac{1}{n} \sum_{i=1}^{n} Z_i^k \tag{2}$$

The $Z$ block value is a linear average of the n point estimators and it has a minimum estimated error variance (Cressie, 2006; Bilonick, 2012). The root mean square error (RMSE) of the kriging is also estimated. This estimate of kriging error is crucial in the Sahelian countries where observation networks are often scantily distributed. We used the kriging RMSE to blank out areas of the country where there are too few rainfall stations to avoid a biased result. We therefore suggest BK as the best available reference candidate.

## 1.2 Satellites and Reanalyses and Combined datasets

To compare the monitoring of dry/wet spells among datasets and their uncertainties, an ensemble of 9 different available datasets were used (See Table 1). These datasets are either satellite products, reanalyses or raingauges. TRMM-3B42 V7 and CMORPH V1.0 are characterized by combining infrared and microwave measurements (Kummerow et al., 1998; Nesbitt et al., 2006; Huffman et al., 2007) while CHIRPS and TAMSAT primarily based on thermal-infrared measurement techniques (Funk et al., 2015; Maidment et al., 2017). Recently, Le Coz and van de Giesen (2019) provide a detailed overview of these products and their recommendations to detect different types of hazards. The infrared measurements are very indirect but they have a high spatiotemporal sampling frequency (Kummerow et al., 1998; Ferraro and Li, 2002; Ferraro, 1997). Conversely, microwave methods enable an improved estimation of instantaneous precipitation but have a low temporal sampling frequency (Joyce et al., 2004; Zeweldi and Gebremichael, 2009; Xie et al., 2017). Thus, in order to benefit from both estimation techniques, the two methods can be coupled, as in TRMM-3B42 V7 and CMORPH V1.0. In addition, the TRMM satellite was the first satellite to have active radar instrumentation on board. As such, it could cover cloud characteristics since the radar was able to measure by means of the principle of electromagnetic wave reflection (Maranan et al., 2018).

Among the datasets, two of them are reanalyses data, namely NCEP-CFSR and ERA5. The NCEP-CFSR is available on the

T382 Gaussian grid (Ebert et al., 2007; Saha et al., 2010), while the ERA5 is based on 4D-Var data assimilation using the 41r2 cycle of the Integrated Forecasting System IFS (Malardel et al., 2016) and is generated by ECMWF. The last dataset, CPC Unified V1.0/RT, is fully based on raingauge observations. It uses the gauge reports of over 30,000 stations worldwide from multiple sources including Global Telecommunication System (GTS), Cooperative Observer Network (COOP) and other

national and international agencies (Xie et al., 2017).

To achieve a reliable comparison and decrease each product's resolution impact, the same spatial/temporal resolutions as the kriging datasets are used. For datasets with a sub-daily temporal resolution, we calculated daily accumulations such as raingauge data. The datasets with spatial resolutions below 0.25 degrees are upscaled to that resolution using bilinear averaging, whereas those with larger spatial resolutions were resampled using bilinear interpolation (Beck et al., 2019).

### 1.3    Methodological Approach

Based on daily rainfall data, different dry/wet spells depending on their duration and intensity are computed for each grid point of the 0.25° shared resolution, over the period of the different data products. The frequency and duration distribution of dry and wet spells depend significantly on the threshold chosen to define a rainy day (Barring et al., 2006). A few authors have used 0.1 mm to define rainy days, as this is the usual accuracy of raingauges (Da et al., 2019). Nevertheless, Frei et al. (2003)

consider 1 mm to be a more relevant measurement for avoiding errors associated with scant precipitation. The authors asserted that precipitation below this amount evaporates directly. In this study, a threshold of 1.0 mm was used. This threshold was also used by Diallo et al. (2016) and Froidurot and Diedhiou (2017) for the Sahel.

This work represents a first step in identifying potential high-impact events. It is therefore important to have a large sample of dry/wet events so as to obtain robust statistics when comparing sources. However, since most of the results presented in

this study concern events with a return period of several years minimum, they could be considered to be extremes or highly abnormal. Moreover, a large number of definitions (related to duration of the episodes and their intensity) are used in order to highlight potential differences between datasets for representing the effect of high-impact events on socio-economic activities. The following subsections present methods applied to detecting dry/wet spells.

### 1.3.1    Dry Spells

Two criteria are used to define two different types of dry spells. The first (hereafter called DS) is based on the number of consecutive dry days having daily precipitation below 1 mm/day. This definition is commonly used to define a dry spell in the Sahel, and the methodology employed here parallels that of Salack et al. (2013). Different DS intensities are defined according to their durations (short DS, medium DS, long DS, and extremely long DS) and are presented in Table 2. The second criterion is based on accumulated precipitation during a specific period, and is called Dry Spell Cumulative (DSC). Four durations (i.e.

intensities) are then defined: DSC5: five days with less than 5 mm of rainfall; DSC10: 10 days with less than 10 mm; DSC15: 15 days with less than 15 mm; and DSC20: 20 days with less than 20 mm (See Table 2). These DSCs can be seen as periods when there is not enough rainfall to significantly moisten the soil and thus not enough for crop growth (Sivakumar, 1992). The

results presented in this study focus on the most intense dry spells (DSC10, DSC20, DSl, DSxl). Nonetheless, all the results from the other periods are presented in supplementary materials.

### 1.3.2 Wet Spells

As for dry spells, two criteria are used to detect wet spells and their intensities. The first method is based on the number of intense rainy days (hereafter called WS). Since intense rainy days may be defined with different relative intensities, four thresholds were also defined ($90^{th}$, $95^{th}$, $99^{th}$, $99.5^{th}$ climatological percentiles of rainy days over all the years and entire seasons). After computing these WS, it was found that durations equal to or longer than 2 days are extremely rare, even for the lowest intensities (percentiles).

Indeed, the wet spell duration categories were chosen to correspond to the different synoptic systems causing rain in West Africa. Short wet spells are associated with the so-called '3–5 day' African Easterly Waves (AEWs). These AEWs are synoptic disturbances known to drive mesoscale convective systems throughout West Africa (Diedhiou et al., 1998; Wu et al., 2013). Because of their wavelengths, only two WS durations were defined: a one-day duration (e.g. to monitor intense daily rainfall, WS1) and equal to or longer than two-day duration (WSM). Thus for example WSM 99P represents a wet event of at least two consecutive days with each cumulated rainfall exceeding the 99th percentile of rainy days. The second criterion in defining wet events is based on percentiles of specific cumulative periods. These cumulative wet spells are defined according to different synoptic components such as the 10-20 day variability mode of African monsoon rainfall which stems from coupled regional land-atmosphere interactions (Grodsky and Carton, 2001; Mounier and Janicot, 2004). Wet spell cumulative (WSC) is defined as specific periods when cumulative rainfall exceeds threshold (as shown in Table 3). As for dry spells, in this study we focused on the strongest wet spells (WS1 99P, WSM 99P, WSC5 99P, WSC15 99P,) although all the results are presented in supplementary materials.

## 2 Results

### 2.1 Seasonal Rainfall over Senegal

The first inter-comparison between all the datasets focuses on total seasonal precipitation from June to October. Fig. 2 shows the climatology of seasonal precipitation on the overlapping dates (1998-2010) between the datasets. The kriging method allows for estimation errors. It takes into account the spatial dependency structure of the data. Based on the kriging error, a critical threshold is established to eliminate pixels when estimated data is not reliable. For this study, the threshold of 0.5 was adopted, based on Lloyd and Atkinson (2001). The main characteristic of Senegalese precipitation, driven by the monsoon flow, is a south-north cumulated rainfall gradient. It is interesting to note that TRMM is closest to BK in intensity but only CMORPH is able to reproduce the specific southeast – northwest gradient observed over the peanut growing basin. This correspondence between TRMM, CMORPH and BK may be due to the precipitation radar (PR) on board the TRMM or the combination of infrared and microwave measurements used in CMORPH since they appear well adapted to this region. Maranan et al. (2018)

report that these instruments provide improved estimation of precipitation by atmospheric assessment of water vapor, cloud water and precipitation intensity.

Nevertheless, the reanalyses appear to underestimate precipitation in the north (ERA5) or south (NCEP) as illustrated Fig. 2f and Fig 2g. The findings show that CMORPH is the product exhibiting the lowest cumulative seasonal rainfall especially in Senegal's southern coastal area compared to other datasets. Indeed in this part of the country, CMORPH records cumulative seasonal rainfall of less than 900 mm, whereas in other datasets rainfall amounts exceed 1,100 mm. This result confirms the findings of Tian et al. (2007) showing that the regular smoothing of precipitation consequential to the "morphing" process can have an effect on precipitation intermittency (Fig. 2c).

There is good correlation among the products in terms of spatial variability and cumulated values (from 100 mm in the north to 1,200 mm in the south), as illustrated in Fig. 3. Using BK as a reference, the performance of the products on the seasonal north-south rainfall gradient is denoted by correlation scores ranging from 0.94 to 0.99. TRMM gives the highest correlation (r = 0.99) and is presented along with OK as the two best-performing products. CMORPH, using the same measurement technique as TRMM, has the lowest score compared to other satellite-based products (r = 0.94). Also, Fig. 3 shows the root mean square error scores which allow us to quantify biases on the intensity of cumulative seasonal rainfall. Satellite products, with the exception of CMORPH, showed the lowest bias with TAMSAT, CHIRPS and TRMM recorded respectively at 56.97, 67.36 and 83.16. Meanwhile, reanalysis and in situ datasets recorded the highest bias.

Fig. 4 showing the cumulative distribution frequency (CDF) of the cumulative seasonal amounts helps to explain these biases. These distributions are calculated on the overlap period between 1998 and 2010. Overall, the products are divided into two groups. First, a group of five products composed of NCEP, ERA5, CPC, OK, and CMORPH where 60% of the cumulative seasonal rainfall is below 600 mm and 90% is below 800 mm. Meanwhile, in the second more heterogeneous group composed of TRMM, TAMSAT, CHIRPS and BK, these thresholds are higher, at 800mm and 1,000mm respectively. This can be explained by the two kriging methods. BK produces an average rainfall estimate at a given location (considered as a "block"). Whereas OK estimates the rainfall value at a point in a region using data near the estimation location. This means the BK method is akin to satellite measurement techniques which also estimate rainfall on pixels (Fig. 3h and Fig. 4h). Finally, there are differences in the peanut growing basin (identified in Fig. 1). This region is an important agricultural area of Senegal supplying 80% of its peanuts for export and 70% of total grain crops, (Thuo et al., 2014). Because of this strategic importance, a consequential network of raingauges (about 24) was used to obtain a more robust estimation of ordinary and block kriging (OK and BK). Regional-scale rainfall patterns are of particular importance. All products showed a similar magnitude of spatial rainfall variations even though this variation is particularly noticeable across the peanut basin with amounts ranging from 400 to 700 mm.

In addition to the cumulative seasonal rainfall, the seasonal progression of the dry days is crucial and is illustrated in Fig. 5. It enables definitions of the start/finish of rainy seasons and drought periods. During the dry season (November to May), all the datasets record over 85% of dry days, showing little occurrence of off-season rainfall called "Heug" rainfall (Seck, 1962; Gaye et al., 1994). Yet, at the same time, OK and BK record 100% of dry days. This could be due to technical issues and/or absence of proper data collection during that period. Given these technical problems, it is even more difficult to declare BK as the most accurate dataset. Nevertheless, certain products such as CMORPH, during the entire dry season, and TAMSAT and NCEP dur-

ing the October to December dry season recorded more "Heug" rainfalls than others datasets (Fig. 5). Although high-intensity rain dominates the wet season, CMORPH misses some of these events in-between scans while overestimating low-intensity events in the dry season. One explanation for this could be the CMORPH's algorithm since it tends to be more sensitive to the false alarm rate (FAR), or the fraction not stemming from events detected by the CMORPH algorithm (Bruster-Flores et al., 2019).

During the rainy season (June to October) when the occurrence of these dry days is more crucial to socio-economic activities, differences between datasets increase, as displayed in Fig. 5. The datasets can be split into groups, TRMM, CPC, NCEP depicting progression close to the OK, with more dry days than the second group (TAMSAT, CHIRPS, CMORPH) through the whole season. The latter is closer to the progression of the BK. Finally ERA5 is the only product similar to OK at the start and end of the season and similar to BK in the middle of it. It is difficult to posit an explanation for the presence of these two groups, CHIRPS and TAMSAT, which combine in situ stations and infrared sensors, and generally record fewer dry days. It is well known that infrared sensors are not well suited to assess ground precipitation from cloud-top temperatures (Ringard, 2017). Another commonality of the two groups is the native resolution of the products. Indeed, even if they are all regridded into the same resolution, TAMSAT, CHIRPS and CMORPH have the highest resolutions (0.0375, 0.05 and 0.05 respectively) compared to TRMM, CPC and NCEP (0.25, 0.5 and 0.31 respectively, see Table 1). Yet this result is counterintuitive since the datasets with coarse resolutions are closer to OK, known to be a point interpolation. Obtaining the largest percentage of dry days via lowest resolutions datasets is very surprising. The seasonal cycle of dry days highlights the complexity of intermittent rainfall in the datasets, and thus the potential difficulty of monitoring dry/wet spells. After this seasonal analysis was carried out, a specific comparison of dry/wet spell detection was done.

## 2.2 Dry Spells

The purpose of this section is to compare the detection of different types of dry spells (depending on their intensity and duration) derived from the 9 products. We will focus on the four most sensitive dry spell indicators for agriculture and livestock, namely DSC10, DSC20, DSl, DSxl (see Table 2 for definitions; further results in supplementary materials). The first comparison concerns the average occurrence of yearly dry spells (Fig. 6). These occurrences are calculated for all datasets only on grid points alone, when the kriging method is considered significant. In Fig. 6 clear differences emerge between BK, TAMSAT and CHIRPS on the one hand, where the number of DSl does not exceed 1, and TRMM, CMORPH, CPC, NCEP, ERA5 and OK on the other hand with average per season recordings of 2 DSl. This pattern persists for DSxl, though there are clear differences for DSC10 and DSC20. This fits with the previous findings concerning dry days (Fig. 5). Such a result confirms the great sensitivity in detecting dry spells using methods which extract precipitation datasets. Indeed, TRMM, with its coupled infrared (IR) and microwave (MW), reports more frequent rainfall breaks than TAMSAT and CHIRPS, which are infrared. Surprisingly, although CMORPH reports finding fewer occurrences of dry days similar to TAMSAT and CHIRPS, it produces comparable occurrences of dry spells to the driest products. This is especially true for DSl and DSxl. TRMM and CMORPH benefit from the advantages of both IR and MO. The IR principle is based on rainfall rate proxy from cloud-top temperatures. According to Dinku et al. (2018) IR sensors might overestimate rainfall rates by considering cirrus clouds to be convective.

Meanwhile, the MW measurement is a more physical measure of clouds' water content, providing a clearer instantaneous estimate of precipitation (Ringard, 2017). This may explain the satisfactory performance of the two products compared to raingauge findings, but does not explain the CMORPH discrepancies between dry day detection and dry spell occurrences. Finally, BK and OK demonstrate important differences in dry spell occurrences. Indeed, the smoothing effect due to kriging is

stronger in OK than BK. This is a direct consequence of the two kriging methods as described above.

To better analyze these different behaviors, seasonal progression is taken into account (Fig. 7) illustrating frequency which is defined as a ratio of observed days having recorded dry spells. Note that, due to their definitions, DSC10, DSC20 and DSxl are quite sensitive to the dry season (November to May), whereas DSl shows rain breaks between 8 and 14 days. Thus, the end of the breaks is necessarily marked by a rainy day, which would explain their sensitivity during the transition phases (i.e.

onset and retreat phase of rainfall) and their misreadings during dry season. Hence, there is a coherent grouping of DSC10 and DSC20 datasets during the rainy season similar to those shown with dry day frequency in Fig. 5. ERA5 and NCEP have an overestimation of DSC10, DSC20 and DSxl during the period from May to July in comparison to other datasets. The monsoon's waning and waxing phases correlate these observations. This detection varies greatly among products. For this particular drought it is difficult to point to the specific behavior of a group of products. Each has a specific time progression

with a higher peak either during onset (June) or retreat phases (September). Some lags are also visible during the retreat phase (TAMSAT for instance). Finally, the very good correspondence between OK and CPC is remarkable to see. Two conclusions can be drawn. First monitoring the seasonal progression of specific dry spells over Senegal is highly complex in spite of alignment on a wider scale (Fig. 7). Secondly, it is difficult to take the reference into account. Even when raingauges are used, it is necessary to spatialize data via kriging methods as this will have a big impact in terms of dry spell detection.

In order to examine the correspondence among the datasets on the spatial variability of the different dry spells, a Taylor diagram (Taylor, 2001) was plotted (Fig. 8). This type of graphic gives an overview of the capacity of datasets to concur on spatial distribution by simultaneously providing three pieces of information: spatial correlation, standard deviation and root mean square deviation (RMSD) compared to a reference. Here, the reference is defined as BK. This is motivated by the fact that this kriging method, with its spatial assessment on grid boxes is more suited for comparison to gridded datasets. Not

surprisingly, DSC20 and DSxl are more stable and thus display the lowest standard deviation values. Spatial correlation is strongest with DSC10, above 0.8 for all datasets, while for the other metrics we find correlations around 0.5 although the dispersion is less marked for DSC20, DSl and DSxl. Overall, TRMM looks to be the closest to the reference, and so the best product for detecting these dry spells. TAMSAT and CHIRPS also get good scores with a correlation above 0.85, standard deviation of 1 and RMSD close to 0.5 for dry spell numbers. In contrast, CPC yields the lowest scores.

It is also worth noting that a big difference stems from the methodology of generating kriging of observation datasets. Hence, OK is generally one of the largest RMSD with the lowest correlation scores. It is important to note that the differences between OK and BK are linked to uncertainties concerning kriging methods for observations. Finally, Fig. 9 depicts a comparison of interannual variability of dry spell occurrences. The figures reveal the challenges of assessing climatological trends due to the high interannual variability of these events, discrepancies between datasets, and sometimes opposing temporal progressions.

Overall, DSC10 and DSC20 display a slight decrease in events. This is noted for all the products except CPC and ERA5.

DSxl displays some similarity to this climatological progression. Nevertheless, interannual variability is much higher, and no significant trend is detected. Finally, DSl denotes a specific time progression. It is worth pointing out that, except for the biases, the time progressions of all the products correspond well, displaying an increase in these DSl at the beginning of the 2000s and peaking in 2003-2004. It is also worth considering that, even if the spatial congruity between the two kriging techniques is low, their interannual progressions are similar.

## 2.3 Wet Spells

In this section, the same intercomparison of datasets monitoring wet spells (depending on intensity and duration) is assessed. In the main document, four types of wet spells using the $99^{th}$ percentile of daily rain amounts as thresholds, namely WS1 99P, WSM 99P, WSC5 99P, WSC15 99P (see Table 3) are discussed. Results using other definitions of wet spells are presented in the supplementary materials. Regarding the intensity of events detected (Fig. 10), there are two main findings. First TRMM appears to be closer to the OK and BK observations than the other datasets. This is true for all wet spell categories. All the other datasets clearly underestimate events, especially when based on OK alone. Regarding BK, which is associated with smoother datasets by definition, there are fewer differences, but they do remain, especially for the WSCs (Fig. 10c,d).

The seasonal cycles of short-duration wet spells (WS1 99P and WSC5 99P, Fig. 11) tend to correspond among the products. As for dry spells this frequency is defined as a ratio of observational days with a recorded wet spell. The only significant differences lie in the underestimation of CHIRPS, CPC and TAMSAT, and CMORPH's delay in representing the peak in the heart of the rainy season. For WSC15, a similar distribution is found and the differences (in terms of intensity or timing) are not huge. Finally, WSM 99P, which is one of the most intense events, displays more variability. It is worth noting that CHIRPS underestimated the WS1 99P and has the most frequent WSM 99P. The reasons for this are not well understood.

In order to elucidate the reasons for these differences, the logarithmic distribution of daily rainfall over the shared 1998 to 2010 period was calculated (Fig. 12). This figure illustrates relatively well how the intensity of daily rainfall can be detected via datasets. Daily rainfalls below 25 mm are more frequent on TAMSAT and CHIRPS. These two products record the most rainy days in the main season (Fig. 2). But the switch to more intense daily rainfall is more abrupt than for the other products. This results in the smallest number of high daily rainfalls for TAMSAT, with a maximum at about 50 mm. CHIRPS and CMORPH are also associated with slight underestimations of strong daily rainfalls (no event above 90 mm). In contrast, TRMM produces the largest rainfall events. This anomaly ranges from mild events (around 30 mm) to the most extreme cases (over 120 mm).

To assess spatial variability of wet spells, the Taylor diagram (Fig. 13) , shows much greater variability than for dry spells (Fig. 8). These results yield globally lower scores for WS than DS due to these events' scarcity and variability. Moreover, differences between products are more pronounced, highlighting the uncertainties of monitoring WS. It is also worth noting that cumulative methods (WSCs) yield better scores. As shown in the previous instance, TRMM appears to be closest to the observations, except for the WSM 99P. This could be due to the very strict criteria for detecting them and the fact that only a few cases were recorded during the shared period. Unusually, despite major discrepancies in daily rainfall distribution (Fig. 12), TAMSAT is relatively well suited to representing events' variability. The fact that our criteria are using quantiles instead of specific rainfall amounts allows such biases to be taken into account with this underestimation. In contrast, CMORPH is generally

the farthest from BK, pointing to the difficulty in representing the spatial variability of these events' occurrences. Finally, recent climatological evolution of these extreme events (Fig. 14) demonstrate a great interannual variability and as for dry spells, important differences among the products. For most of the products the temporal correlations are not significant when compared to BK or OK. Recently, for almost all the products and wet spells, there was an increase in occurrences, especially for WSC5 99P and WSC15 99P. For the two products based on observations, the temporal progressions are quite close and display a major increase in all indicators. These results are in line with the recent study by Taylor et al. (2017) suggesting that Mesoscale Convective Systems (MCSs) responsible for extreme rainfall in the Sahel increased recently. However, this could also be related to a highly abnormal year in 2010.

## 3    Discussion

In this study, a wide range of datasets are compared to assess uncertainties in monitoring dry/wet spells in Senegal. Significant differences and discrepancies are observed. The product resulting from the BK method of in situ observations is identified as a reference. This is justified by the fact that krigged data are more likely to be comparable to gridded satellite observations or model data. This method, representing mean precipitations on a grid, is also more comparable to integrated data from other products.

The first investigation considered the resolution of the products. Even if all the products are regridded on identical grids, the original resolution of the products differs greatly from one product to the next. Disparities between OK and BK, even though they come from the same raingauge networks, are akin to the differences between the two kriging methods. Indeed, the OK method is used to estimate a value at a point in a region, applying data close to the estimation point, while the BK method uses a movable zone or block. Therefore, the smoothing effect resulting from kriging is stronger in BK than OK, since it tends to diminish rain event intensity and augment rainy day occurrences. However, the results obtained were counter-intuitive, especially for dry spells, with more dry spells coming from the lowest resolution datasets. For wet spells, it turns out that products with the lowest intense rainfall are also the highest resolution datasets. Therefore, the resolution of datasets is probably an insufficient explanation for these differences. Furthermore, satellite products combining infrared and microwave result in good sampling (IR) with improved intensity extractions (MO). TRMM and CMORPH using this combination show similar skill in detection of wet spell intensity and are often quite close to in situ observations. Moreover, the correspondence between TRMM and raingauges (OK and BK) seems to point to the importance of the contribution of radar on board the TRMM satellite. It should be remembered that TRMM was the first satellite to be equipped with an active radar instrument on board. This represents great added value since it provides a profile of rainfall activity. It is also important because the data obtained indicate in-cloud precipitation structure, type and vertical extent of this precipitation, and freezing point height determined via bright band level. As far as the reanalyses (ERA5 and NCEP) are concerned, they quickly reach their limits in reproducing these precipitation events. It is important to note that precipitation is generally not a reanalysis product but is rather derived from short-term forecasts in the reanalysis cycle. Observations are thus not assimilated and the products are generally seen by the providers as less robust. Overall, these results confirm the conclusions of Siegmund et al. (2015) on reanalyses. Indeed, in

unimodal regions such as the Sahel, where a unique rainy season is observed from June to October, the reanalyses are quite close to the main characteristics of monthly and annual rainfall. This contrasts with the Gulf of Guinea regions where there is a bimodal rainfall regime. However, reanalyses often show quite significant differences in intra-seasonal rainfall characteristics.

## 4  Conclusions

In this work, the monitoring of high-potential impact events over Senegal is studied using 4 satellite products (TRMM 3B42 V7, CMORPH V1.0, CHIRPS V2.0, TAMSAT V3), 2 reanalyses models (NCEP-CFSR, ERA5) and 3 raingauge-based observations (CPC Unified V1.0/RT, OK, BK). For this, the same spatial resolution was applied to all the products via area averaging, interpolation or kriging to obtain a single spatial resolution of 0.25 x 0.25 degrees. Large-scale climatology research on seasonal rainfall points to decent correspondence between the products, particularly for the well-known south-north rain-

fall gradient associated with the West African monsoon. Some differences in the magnitude of seasonal rainfall amounts are observed however when pinpointing on specific regions. TRMM, TAMSAT and CHIRPS yield seasonal cumulated rainfall quite close to BK. This specific kriging technique is chosen as the reference since its estimation covers average rainfall over pixels, similar to most satellite products. Nevertheless, this similarity among products lessens when analyzing the seasonal cycle of dry days. Two data groups emerge, one recording more dry days with less correspondence between different data

products for dry spells vs. wet spells. Indeed, especially for WS, TRMM and CMORPH, they are quite close to OK and BK. This correspondence illustrates the value of combined IR and MO techniques that optimize the advantages and shortcomings of both types of remote sensing. Nevertheless on WSC, TRMM maintains its correspondence to OK and BK, unlike CMORPH which tends to be closer to ERA5 and NCEP. The TRMM on-board radar appears to play an important role because of its close correspondence to raingauges, especially in WS and WSC. Moreover, the WS intensities in TRMM, OK and BK are often

more than double those of TAMSAT and CHIRPS. This exemplifies the difficulties of satellite datasets which use only infrared sensors. The reason for this is that cold but non-precipitating cirrus clouds impact the infrared with very cold temperatures, so the system sees these clouds as precipitating. Finally, interannual progressions of dry/wet spells were compared. We noted a slight trend toward DS decrease for the products as well as a positive but non-significant WS trend. This insignificance may be explained by the extremely short durations of the products available.

This study shows that despite the general correlation on seasonal precipitation, there is extensive uncertainty about monitoring extreme dry/wet spells on an intra-seasonal timescale. Nevertheless, since there is a marked proximity between TRMM and raingauges for all dry/wet spell categories, TRMM may be a prime candidate for extrapolating these results to other areas of West Africa. Our study reveals several potentially important implications, in particular concerning the judicious choice of datasets to implement early warning systems (EWS) integrating a multi-hazard approach and disaster risk management plus

adaptation to a "hydroclimatic intensity" context. This study also provides useful information for different hydrological and agronomic applications by defining a wide range of rainfall metrics. This may benefit agricultural insurance companies as well as stakeholders by implementing more effective indicators for considerably-improved mitigation measures.

*Author contributions.* NF made the analysis and, NF, CL, MD GP discuss the results and wrote the paper. GP supports this study for the kriging methods. AG advises and provides scientific recommendations

*Competing interests.* The authors declare that they have no conflict of interest.

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

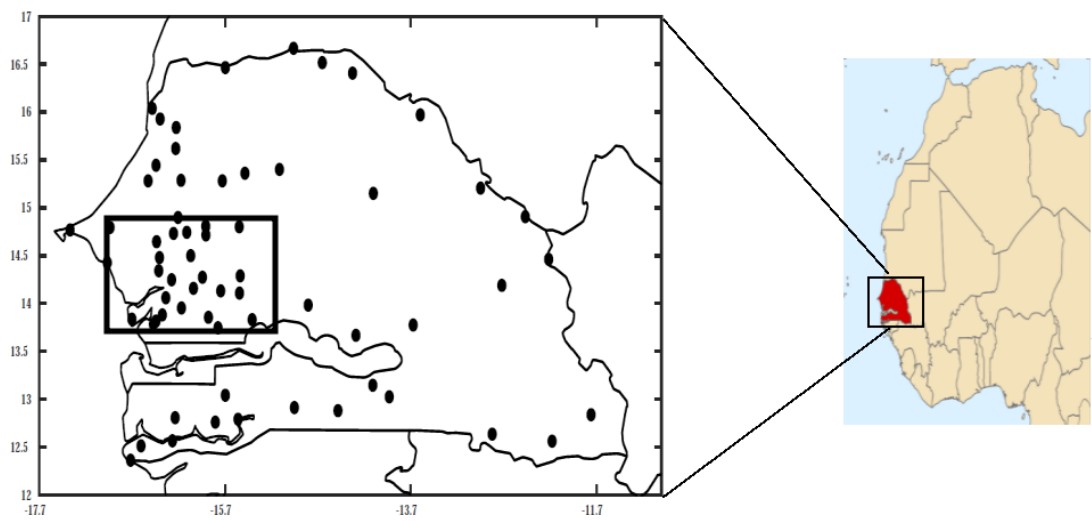

**Figure 1.** Map of Senegal and West Africa (inset). The black dots indicate the location of the 65 ANACIM raingauges used in this study. The square in central western Senegal denotes the location of the peanut basin (area of high density of raingauges).

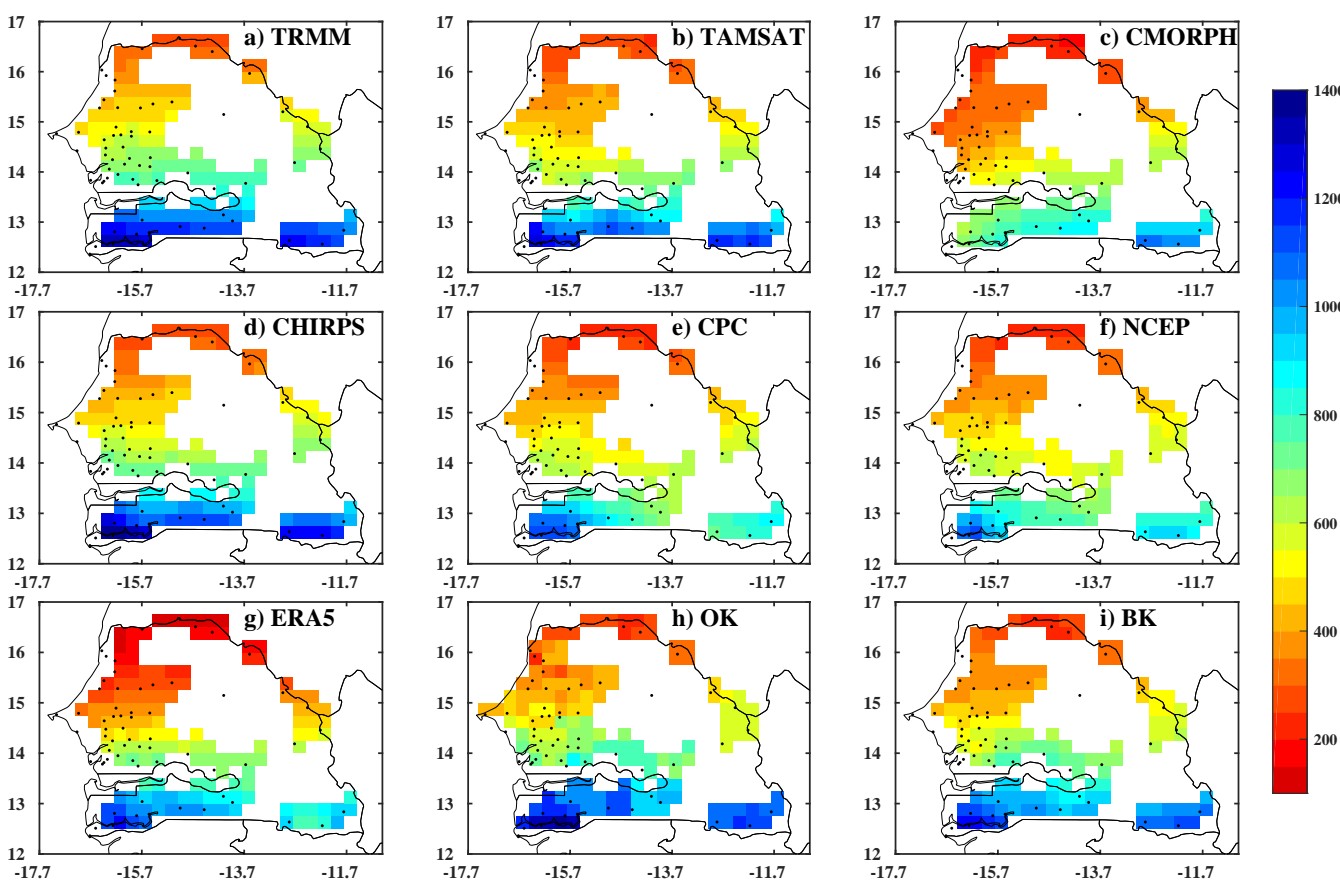

**Figure 2.** Spatial distribution of average seasonal rainfall from June to October on the overlap period between datasets (1998-2010, in mm): using a) TRMM b) TAMSAT c) CMORPH d) CHIRPS, e) CPC f) NCEP g) ERA5 h) OK i) BK. The black dots represent the stations used. Details of the datasets are provided in Table 1

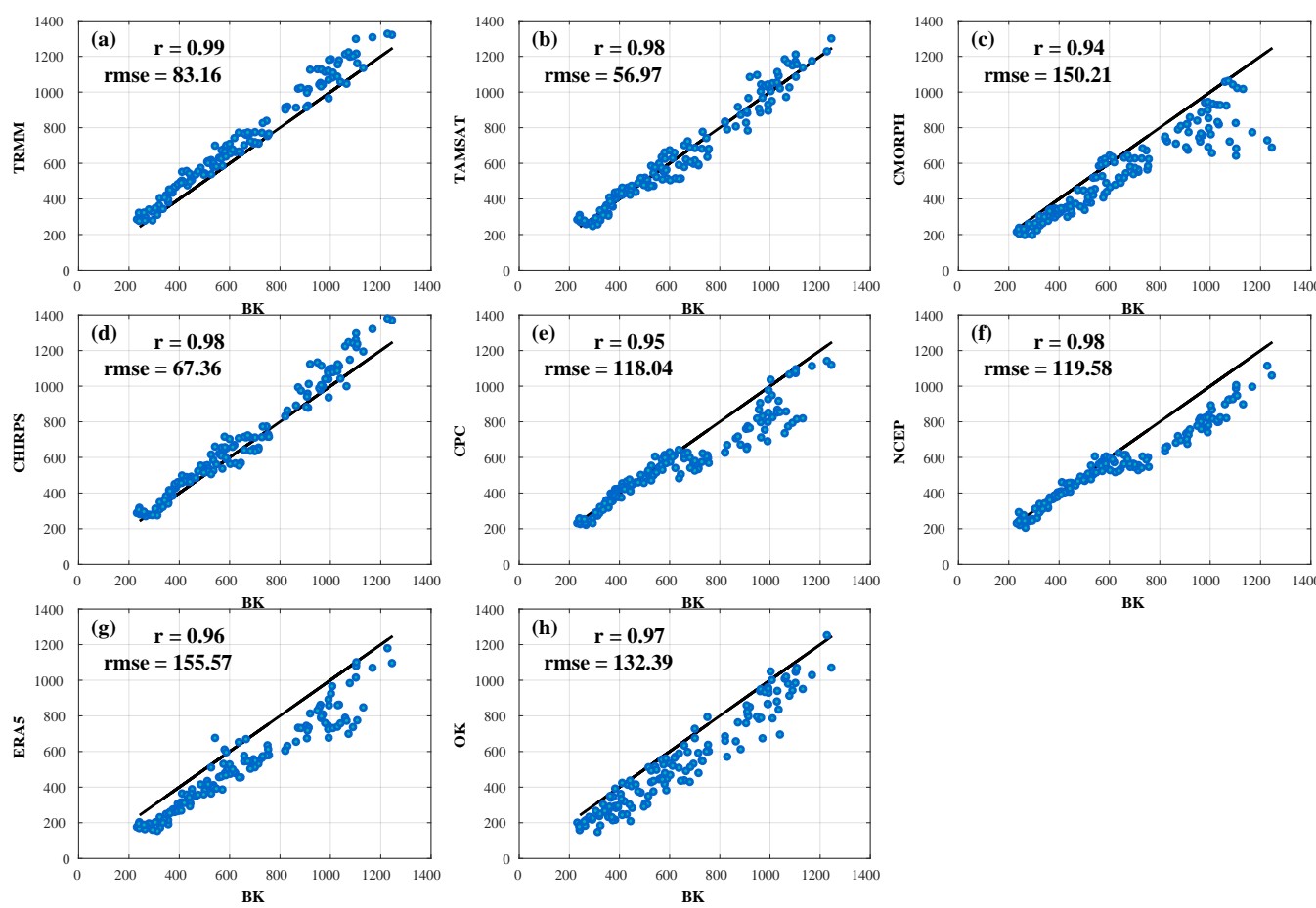

**Figure 3.** Scatter plots of cumulative seasonal rainfall from rain-gauge BK versus a) TRMM b) TAMSAT c) CMORPH d) CHIRPS e) CPC f) NCEP g) ERA5 h) OK. The RMSE and correlation scores are spatial and computed for the rainy season (June to October) over the 1998-2010 period. Details of the datasets are provided in Table 1

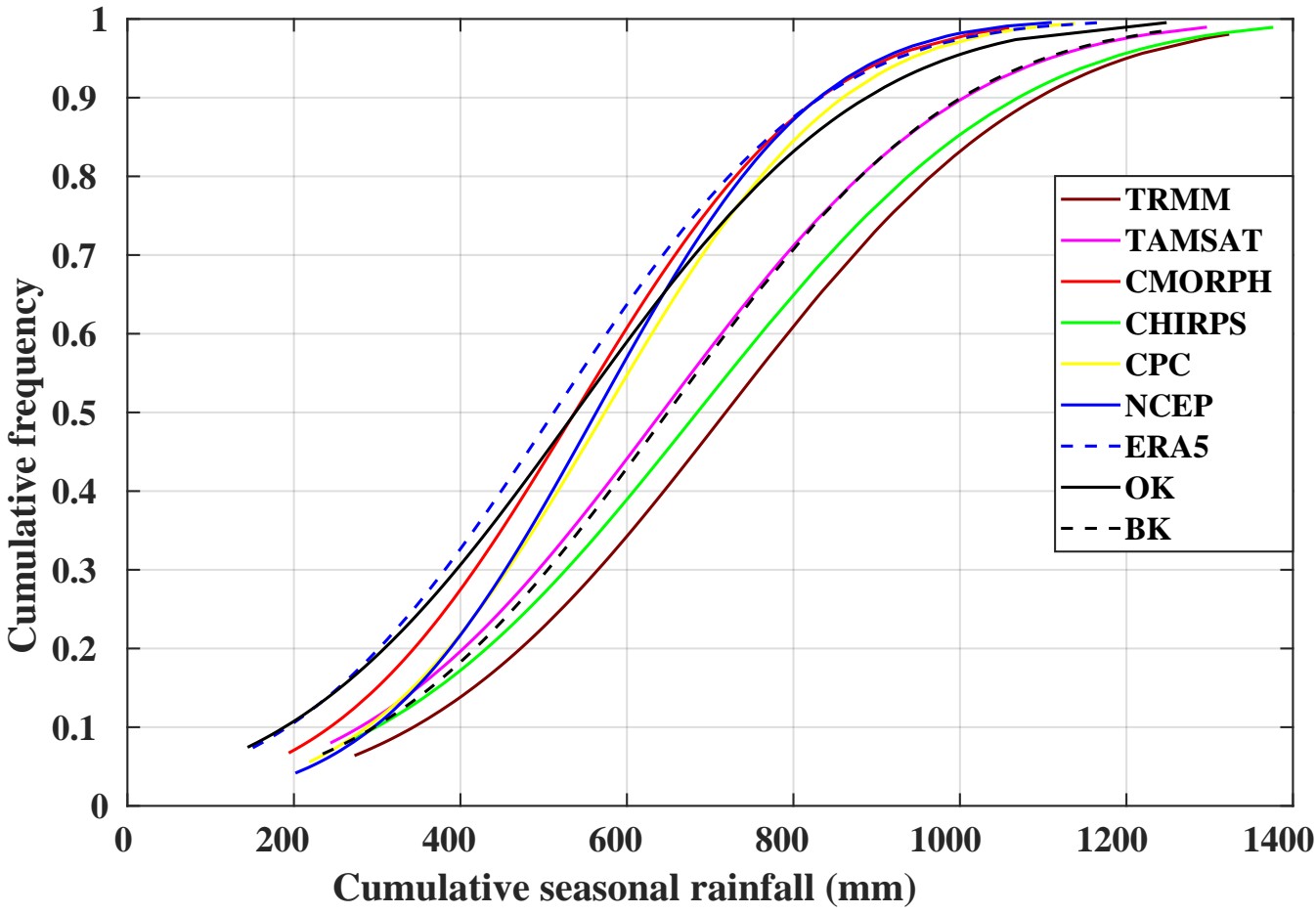

**Figure 4.** Distribution of cumulative density frequency (CDF) of amounts of the seasonal rainfall from June to October on the overlap period between datasets (1998-2010, in mm): using TRMM, TAMSAT, CMORPH, CHIRPS, CPC, NCEP, ERA5, OK and BK. Details of the datasets are provided in Table 1

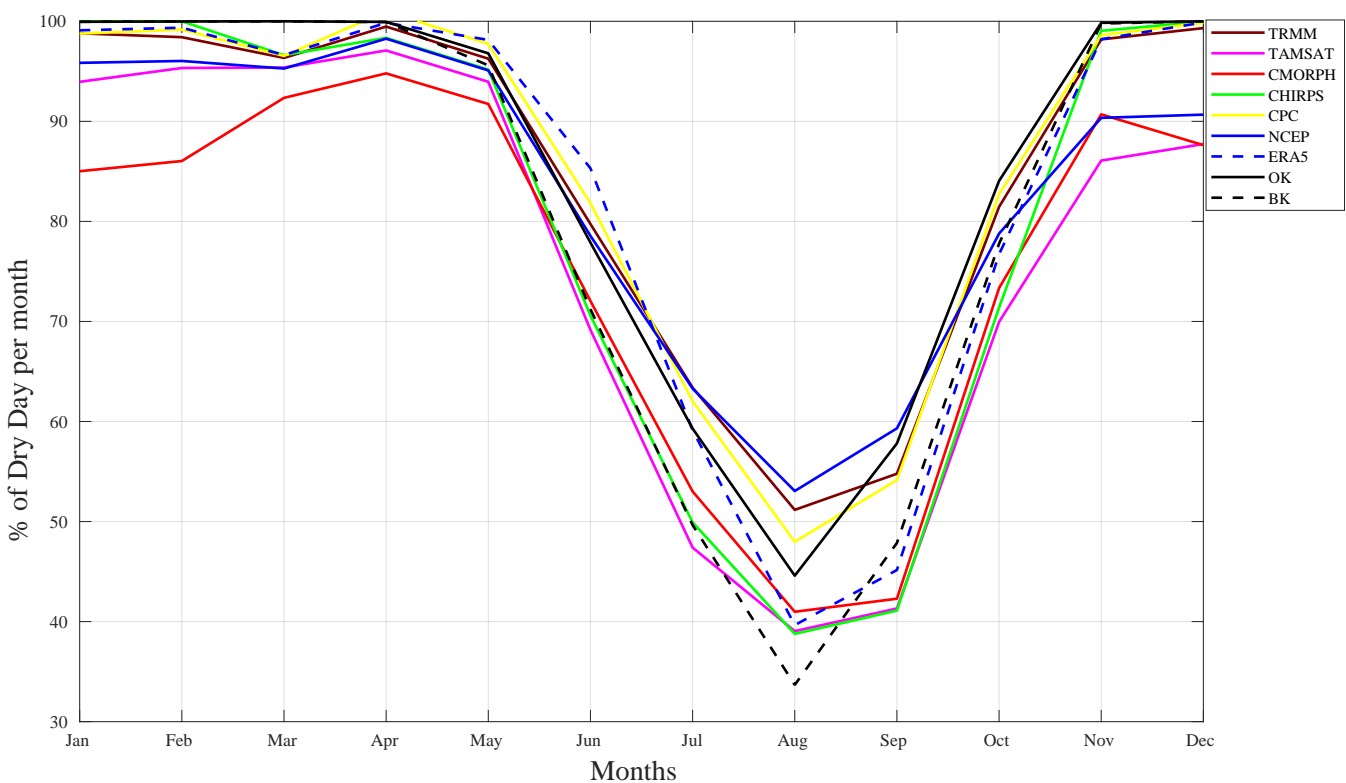

**Figure 5.** Percentage of average dry days (<= 1 mm) per month computed on the overlap period between datasets (1998-2010) and for all grid points in each dataset: TRMM, TAMSAT, CMORPH, CHIRPS, CPC, NCEP, ERA5, BK, OK.

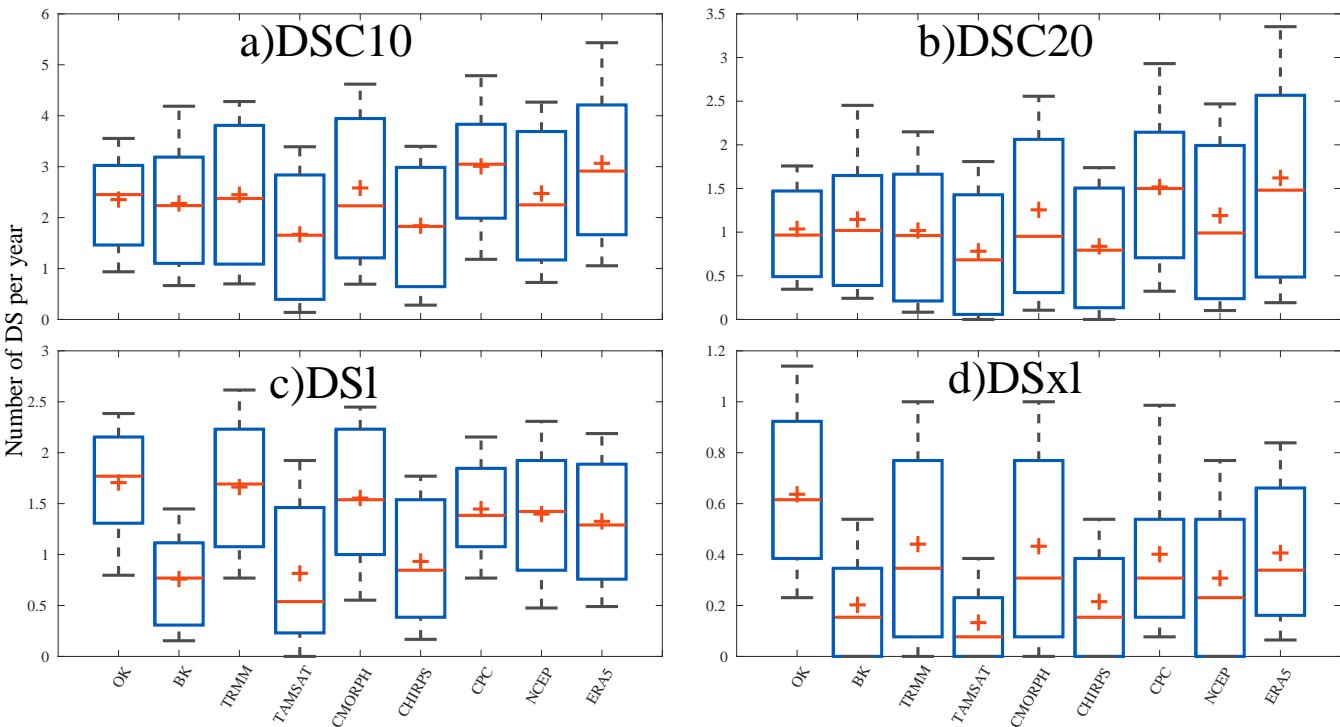

**Figure 6.** Box plots of average number of dry spells (DSC10, DSC20, DSl and DSxl) per year collected on all grid points for the 9 gridded datasets used ( TRMM, TAMSAT, CMORPH, CHIRPS, CPC, NCEP, ERA5, BK, OK). The minus sign (-) represents the median value, the plus sign (+) represents the mean value, the bottom and top edges of the box represent the 25th and 75th percentile values, respectively, while the "whiskers" represent the extreme values (5 and 95%). The average number of dry spells is computed on the overlap period (1998-2010). Details on the datasets and dry spells are provided in Tables 1 and 2, respectively.

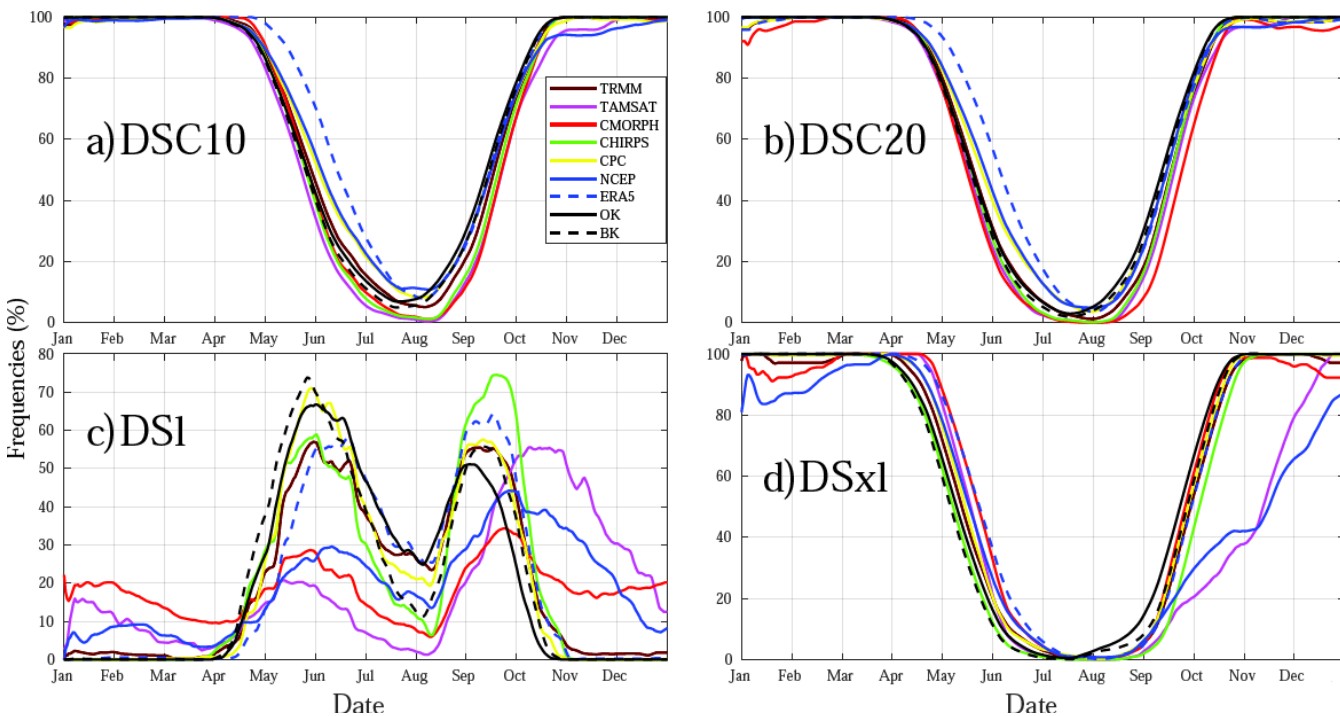

**Figure 7.** Seasonal cycle of four categories of dry spells (DSC10, DSC20, DSl and DSxl) used in this study computed on the overlap between datasets (1998-2010): TRMM, TAMSAT, CMORPH, CHIRPS, CPC, NCEP, ERA5, BK, OK. Frequency is defined as a ratio of observational days with recorded dry spells. Details on the datasets and dry spells are provided in Tables 1 and 2, respectively.

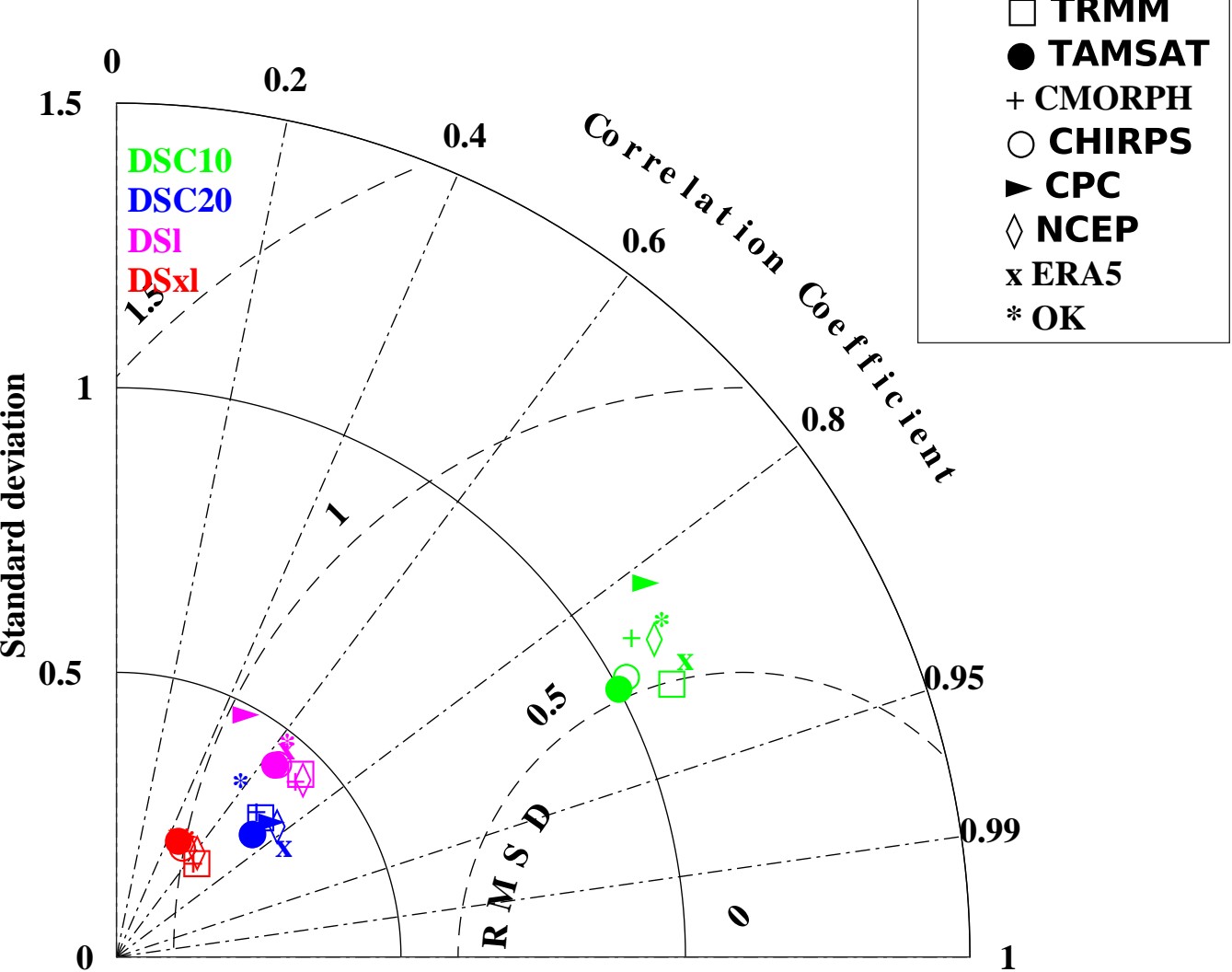

**Figure 8.** Taylor diagram providing 3 statistical scores (standard deviation, correlation coefficient, root mean square deviation), where radius expresses the standard deviation, the angle the correlation, and the distance from the bottom right point the RMSD. The BK dataset is considered as a reference for comparing the spatial distribution of the four categories of dry spells (DSC10, DSC20, DSl and DSxl) of the different datasets (TRMM, TAMSAT, CMORPH, CHIRPS, CPC, NCEP, ERA5, OK). BK is used as a reference. Details on the datasets and dry spells are provided in Tables 1 and 2, respectively.

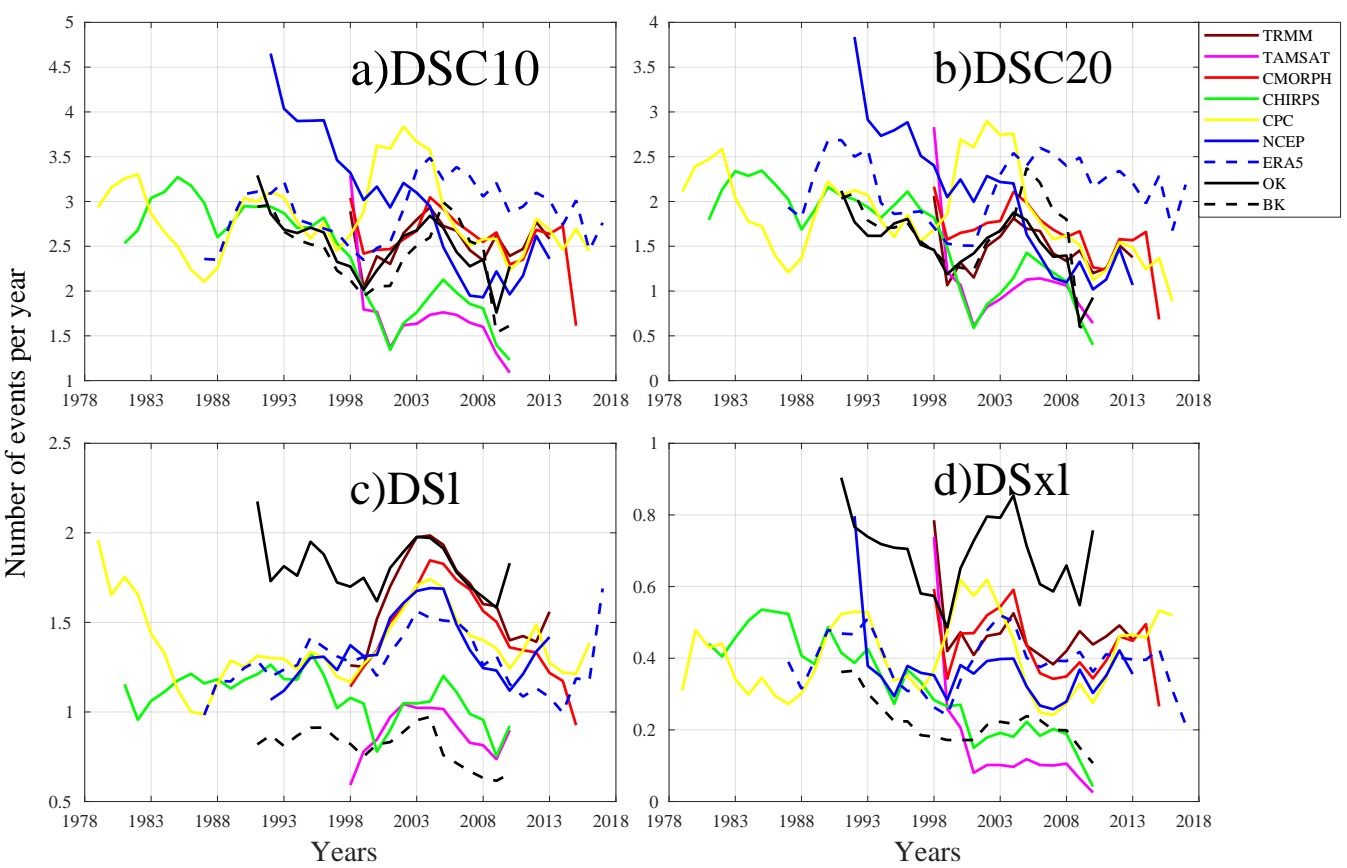

**Figure 9.** Interannual variability of average numbers on all grid points of DSC10, DSC20, DSl and DSxl computed over the period of availability for each dataset: TRMM (1998-2013), TAMSAT(1998-2010), CMORPH (1998-2015), CHIRPS (1981-2010), CPC (1979-2016), NCEP (1992-2013), ERA5 (1987-2017), BK (1991-2010), OK (1991-2010). Details on the datasets and dry spells are provided in Tables 1 and 2, respectively.

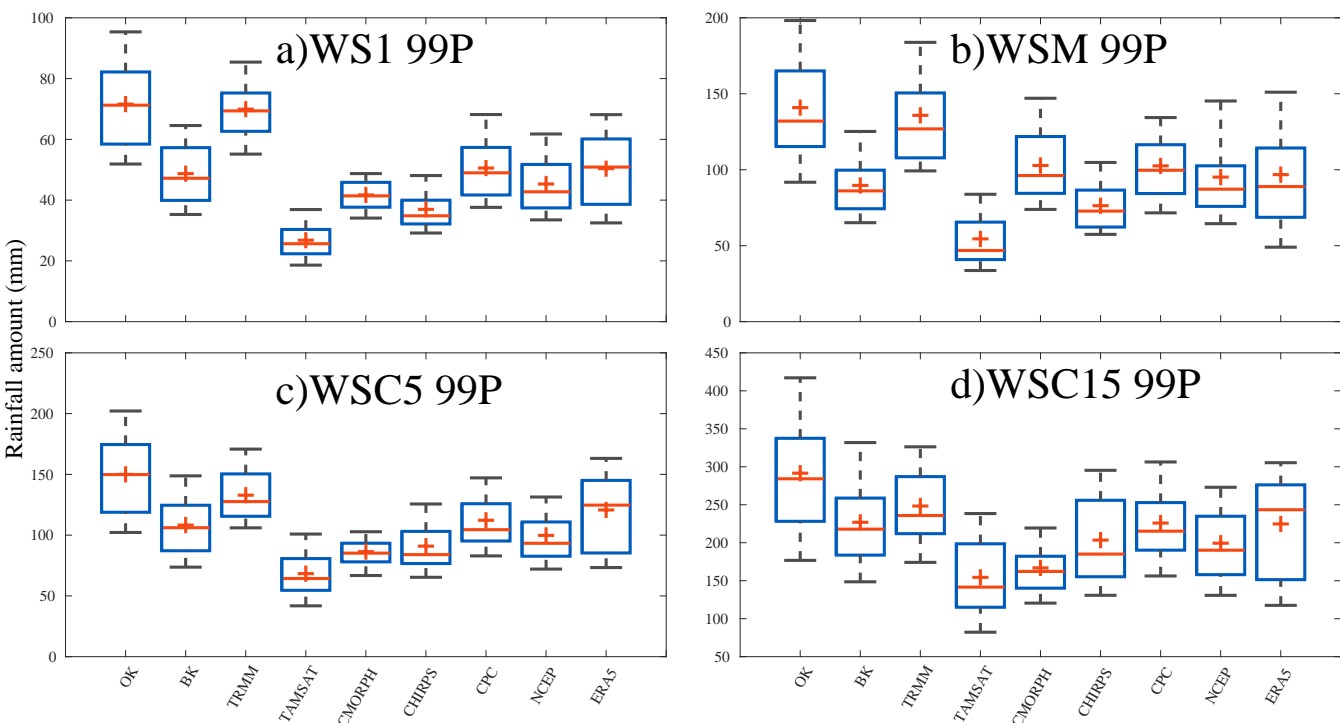

**Figure 10.** Box plots of amounts on all grids points from wet spells WS1, WSM, WSC5 and WSC15 (99th), per year, collected on all grid points for the 9 gridded datasets used ( TRMM, TAMSAT, CMORPH, CHIRPS, CPC, NCEP, ERA5, BK, OK). The minus sign (-) represents the median value, The plus sign (+) represents the mean value, the bottom and top edges of the box represent the 25th and 75th percentile values, respectively, while the "whiskers" represent the extreme values (5 and 95%). The average number of wet spells is computed on the overlap period (1998-2010). Details on the datasets and wet spells are provided in Tables 1 and 3, respectively.

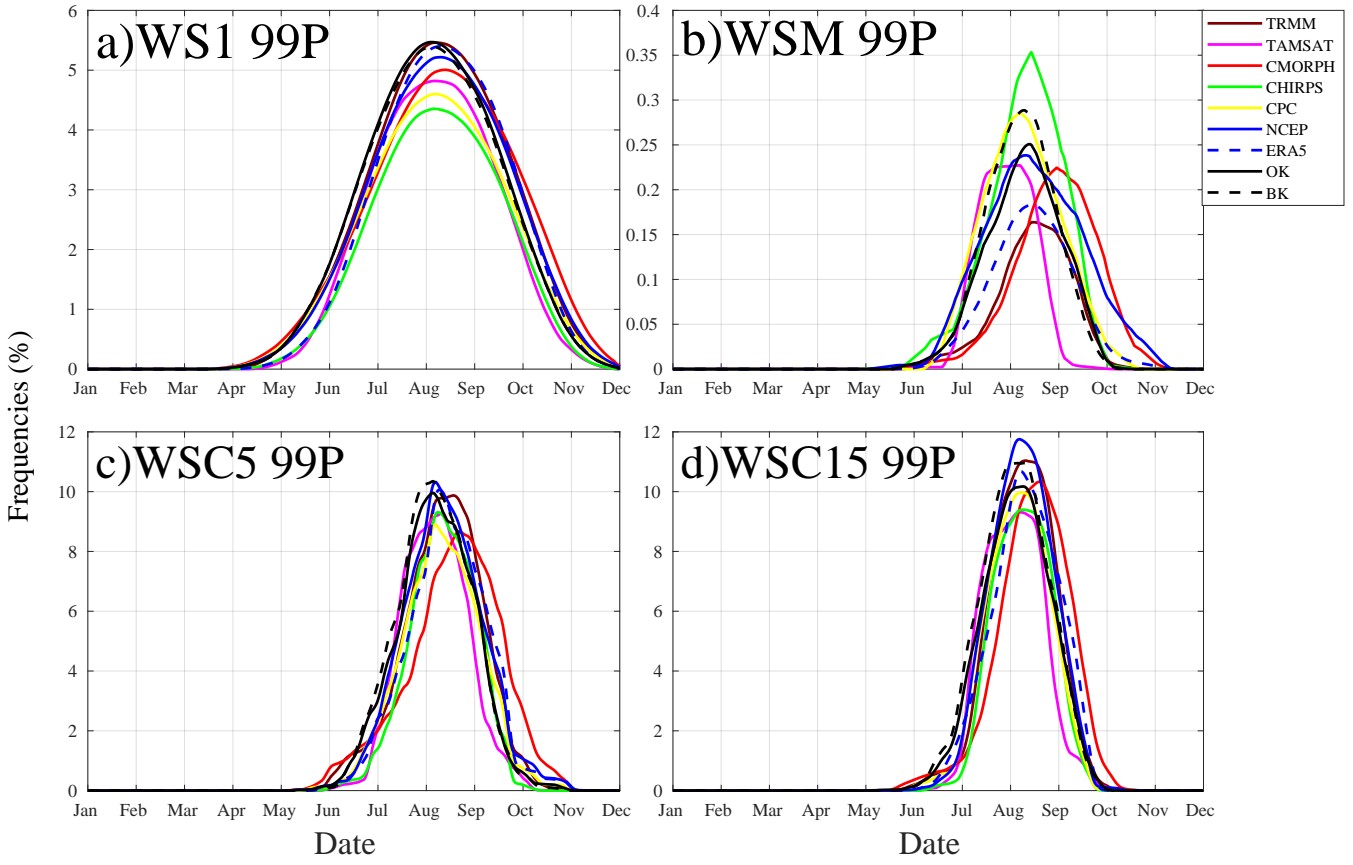

**Figure 11.** Seasonal cycle of four categories of wet spells WS1, WSM, WSC5 and WSC15 ($99^{th}$ percentile) used in this study computed on the overlap between datasets (1998-2010): TRMM, TAMSAT, CMORPH, CHIRPS, CPC, NCEP, ERA5, BK, OK. Frequency is defined as a ratio of observational days with recorded wet spells. Details on the datasets and wet spells are provided in Tables 1 and 3, respectively.

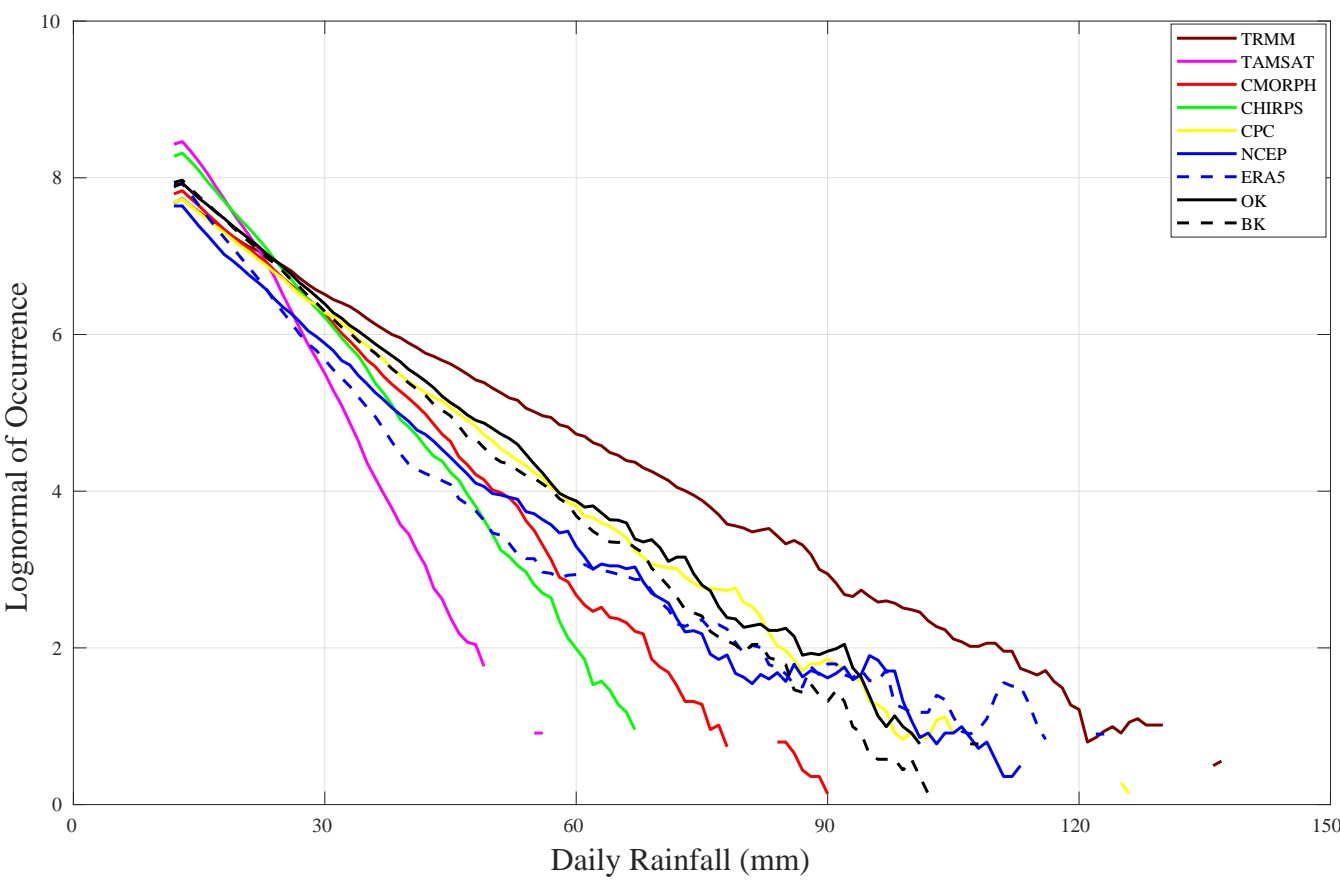

**Figure 12.** Comparison of the logarithmic distribution of daily rainfall amounts recorded at each grid point over the common period between datasets for Senegal (1998-2010): TRMM, TAMSAT, CMORPH, CHIRPS, CPC, NCEP, ERA5, BK, OK . Details of the datasets are provided in Table 1

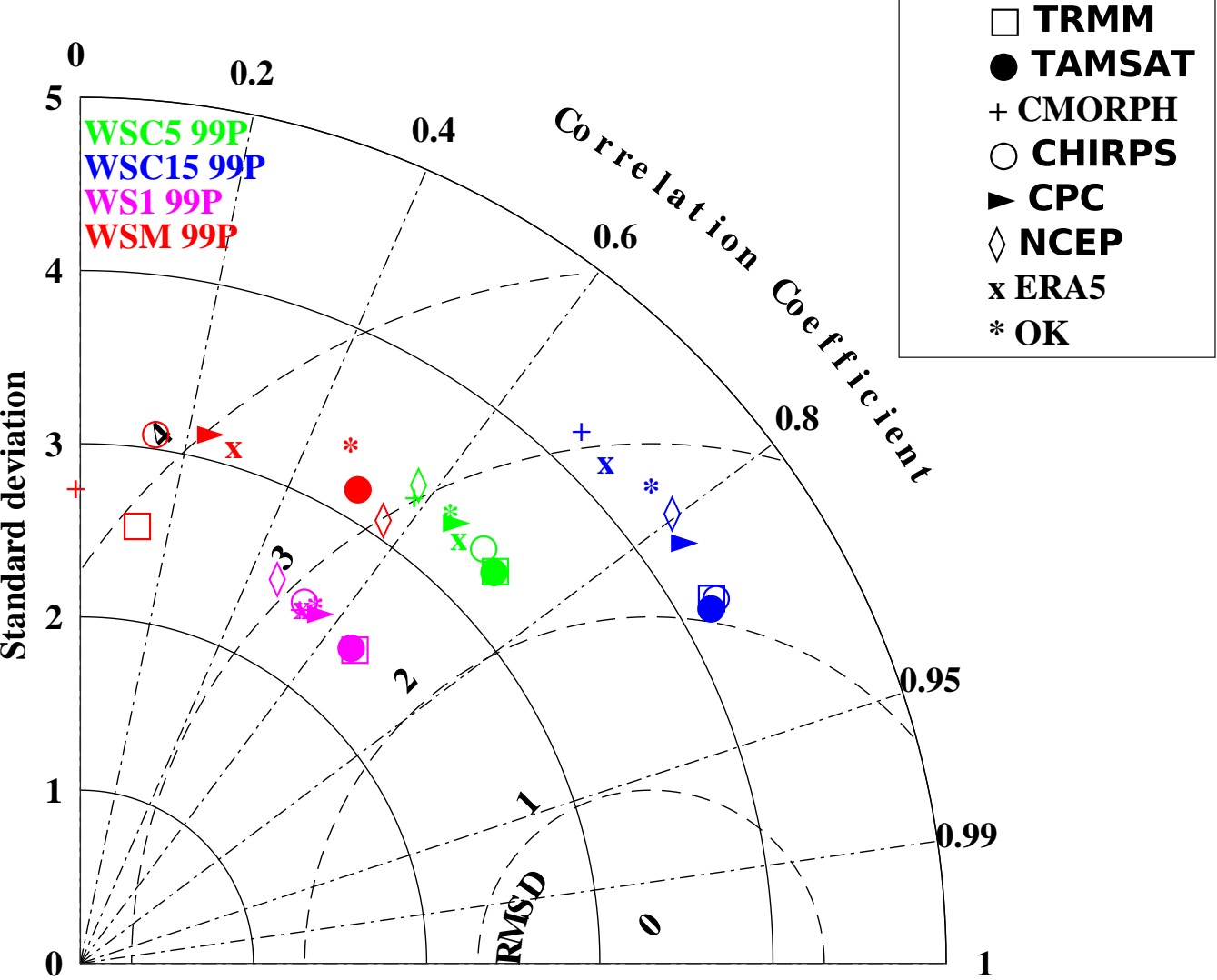

**Figure 13.** Taylor diagram providing 3 statistical scores (standard deviation, correlation coefficient, root mean square deviation) where radius expresses the standard deviation, the angle the correlation, and the distance from the bottom right point the RMSD. The OK dataset is considered as a reference for comparing the spatial distribution of the four categories of wet spells WS1, WSM, WSC5 and WSC15 (99th) of the different datasets (TRMM, TAMSAT, CMORPH, CHIRPS, CPC, NCEP, ERA5, OK). BK is used as a reference. Details on the datasets and wet spells are provided in Tables 1 and 3, respectively.

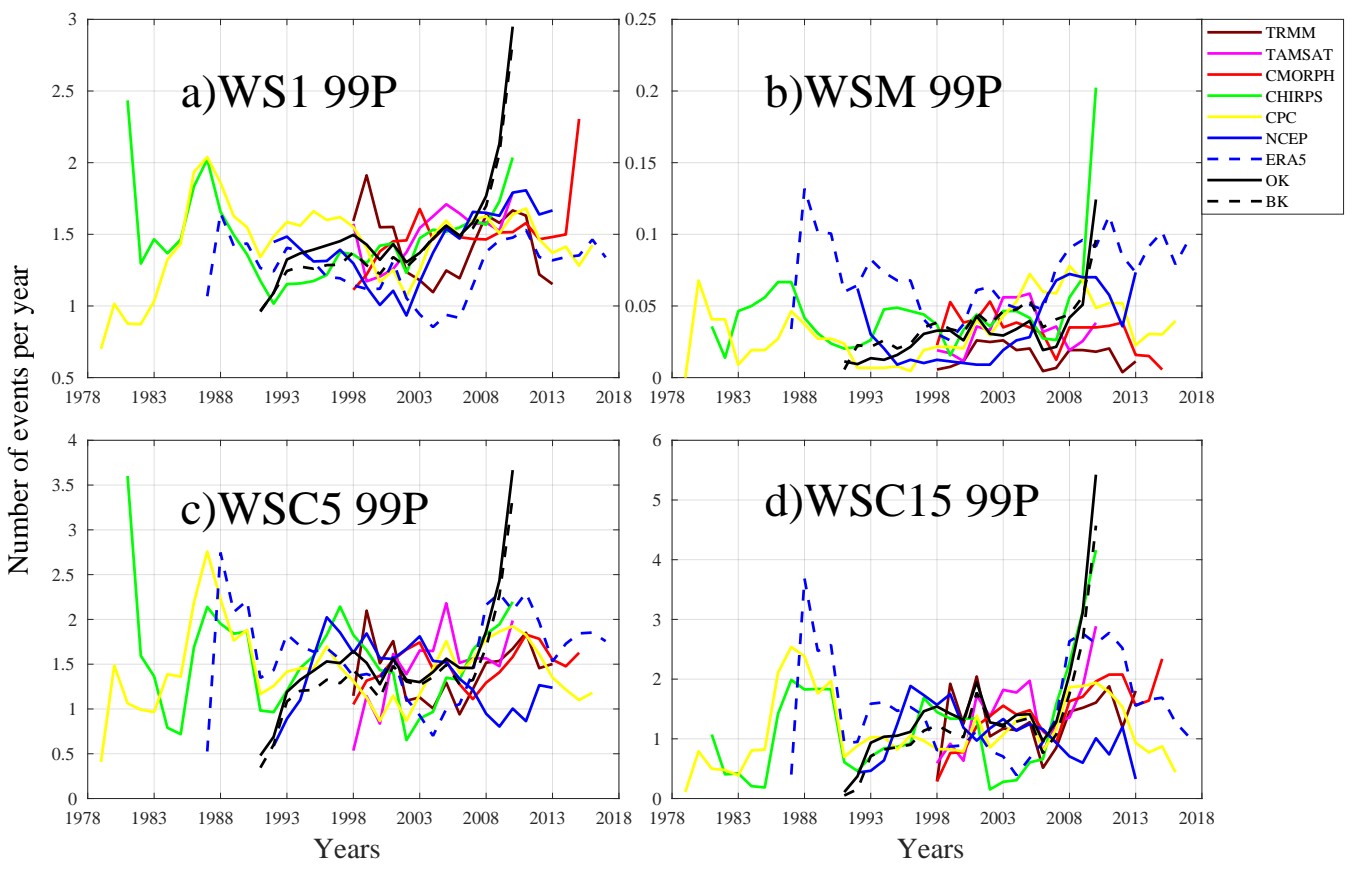

**Figure 14.** Interannual variability of average numbers on all grid points of WS1, WSM, WSC5 and WSC15 ($99^{th}$) computed over the period of availability for each dataset: TRMM (1998-2013), TAMSAT(1998-2010), CMORPH (1998-2015), CHIRPS (1981-2010), CPC (1979-2016), NCEP (1992-2013), ERA5 (1987-2017), BK (1991-2010), OK (1991-2010). Details on the datasets and wet spells are provided in Tables 1 and 3, respectively.

**Table 1.** Summary of the 9 datasets used in this study. The abbreviations in the data sources column are defined as follows: S = satellite R = reanalysis; G = raingauge.

| Name | Details | Data Sources | Spatial Resolution | Temporal Resolution | Temporal Coverage |
|---|---|---|---|---|---|
| TRMM 3B42 (V7) | Tropical Rainfall Measuring Mission (TRMM) 3B42 (V7) | S | $0.25^o \times 0.25^o$ | $3 - hourly$ | 1998 - $present$ |
| CMORPH V1.0 | CPC MORPHing technique (CMORPH) V1.0 | S | $0.05^o \times 0.05^o$ | $30min$ | 1998 - $present$ |
| CHIRPS V2.0 | Climate Hazards group InfraRed Precipitation (CHIRP) V2.0 | S, R, G | $0.05^o \times 0.05^o$ | $Daily$ | 1981 - $present$ |
| TAMSAT V3 | Tropical Applications of Meteorology using satellite data and ground based observations V3 | S, G | $0.0375^o \times 0.0375^o$ | $Daily$ | 1983 - $present$ |
| NCEP-CFSR | National Centers (CFSR) for Environmental Prediction (NCEP) Climate Forecast System Reanalysis | R | $0.31^o \times 0.31^o$ | $Hourly$ | 1979 - 2010 |
| ERA5 | European Centre for Medium range Weather Forecasts ReAnalysis 5 (ERA5) | R | $0.25^o \times 0.25^o$ Native resolution is 9 $km$ | $Hourly$ | 1987 - $present$ |
| CPC Unified V1.0/RT) | CPC Unified Gauge-based Analysis of Global Daily Precipitation V1.0/RT | G | $0.5^o \times 0.5^o$ | $Daily$ | 1979 - $present$ |
| OK | Ordinary Kriging | G | $0.25^o \times 0.25^o$ | $Daily$ | 1991 - 2010 |
| BK | Block Kriging | G | $0.25^o \times 0.25^o$ | $Daily$ | 1991 - 2010 |

**Table 2.** Definition of indexes for detecting dry spells

| Dry Spells Indices | Definitions |
| --- | --- |
| DSC5 | 5 days with less than 5 mm of rainfall |
| DSC10 | 10 days with less than 10 mm of rainfall |
| DSC15 | 15 days with less than 15 mm of rainfall |
| DSC20 | 20 days with less than 20 mm of rainfall |
| DSs | 1-3 consecutive dry days |
| DSm | 4-7 consecutive dry days |
| DSl | 8-14 consecutive dry days |
| DSxl | consecutive dry days exceding 15 days |

**Table 3.** Definition of indexes for detecting wet spells, XX for the 90, 95, 99 and 99.5 percentiles

| Wet Spells Indices | Definitions |
| --- | --- |
| WS1 *XX*P | 1day with rainfall $> XX^{th}$p of daily rainfall |
| WSM *XX*P | 2 day or more with rainfall $> XX^{th}$p of daily rainfall |
| WSC5 *XX*P | 5-d precip. $> XX^{th}$p of 5-day cumulative rainfall |
| WSC10 *XX*P | 10-d precip. $> XX^{th}$p of 10-day cumulative rainfall |
| WSC15 *XX*P | 15-d precip. $> XX^{th}$p of 15-day cumulative rainfall |
| WSC20 *XX*P | 20-d precip. $> XX^{th}$p of 20-day cumulative rainfall |