# Peer review of "Wet and dry spells in Senegal: Comparison of detection based on satellite products, reanalysis and in-situ estimates"

_Natural Hazards and Earth System Sciences, 2019_

## Referee Comment (RC1) · Anonymous Referee #1 · 15 Sep 2019

General comments

This paper analyses the occurrence of wet and dry spells in Senegal estimated from a set of precipitation products (in situ, satellite and reanalysis). Although the purpose of the paper is valuable and results potentially significant, the overall quality of the analysis and presentation is below acceptable standard.

1) Language needs substantial revision and improvement, by a native speaker if possible;

2) The presentation of the method (description of kriging in Section 1.1 and definition of Wet and Dry spells in Section 1.3) needs improvement, the description of the methods

should be expanded and the presentation clarifyed;

3) The presentation of the results is often unclear, and results themselves are not discussed enough: Section 2 is basically a description of the figures, and the key-point of the paper, i.e. the comparison of the precipitation datasets, is never actually addressed;

4) Figure captions should be improved;

5) Conclusion section is not conclusive at all, it is just a summary of the paper.

For the above reasons I cannot recommend the publication of the paper in NHESS. I suggest the authors to undertake a substantial revision of the paper, and resubmit it for new consideration. I list below my specific comments that I hope can help the authors in improving the paper.

Specific comments

Page 1, line 6: "more variability", do you refer to some intra-dataset variability? If this is the case, this expression is not suitable, please rephrase.

Page 1, lines 9-10: these lines are unclear.

Page 1, line 15: add the reference to Taylor et al. 2017, https://www.nature.com/articles/nature22069.

Page 1, lines 15-16: this sentence should be moved at line 19, before "Recently extreme events..."

Page 2, lines 25-27: sentence unclear.

Page 2, line 30: when stating the objective of the paper, it is not clear that you compare satellite/reanalysis products to rain gauges, which you consider as reference. Please clarify this point.

Page 2, line 33: are wet and dry spells "extreme hazards"?

Page 3, line 1: DS, DSC, WS and WSC are not defined in the text.

Section 1.1: do rain-gauge time series have passed any objective homogeneity check?

Section 1.1: BK is poorly described.

Equation 1: how the lambda weight are assigned?

Page 3, line 18: why kriging reduces high values and increase low values? Please explain. Equation 2: what is xq?

Page 3, line 22: square root of OK variance is used for what?

Section 1.2: there are discrepancies in dataset resolutions in the text and Table 1, please check.

Page 5, line 9: where did you define wet and dry spells as extremes? "A maximum number of definition", this sentence is unclear.

Page 5, line 16: on which statistical basis do you define "DS extreme long"? You should use the "extreme" word carefully, and only after an analysis of distributions.

Page 5, line 20: what do you mean by "DSC duration is known"? Please clarify.

Page 6, line 12: why BK is more comparable to averaged values and more in agreement with satellites?

Page 6, line 14: please explain why radar on board of TRMM explain TRMM performance.

The TRMM satellite is the first satellite with an active radar instrument onboard. It is a powerful added value since it provides a profil of the rainfall activity. This is especially important over tropical region where unsateurated downdraft and evaporation of the rainfall is important. It is also important since the rainfall estimates is not based on a proxy of the top of the convective cells. This estimation is quite common and derived from passive radiometer but shows some bias (overestimation) during the collapsing period of the convective cells. This point is now clarified in the document.

Page 6, line 15: please improve the description of the Peanut Basin or highlight it on the map, to facilitate the reader to locate the region.

Page 6, lines 20-21: This actually means that during dry season date are not collected, isn't it? In this case you cannot consider these days as dry.

This is correct. We have clarified the document as suggested.

Page 6, line 26: does "variability" refer to the datasets? I suggest to find another

expression in this case. How can I compare seasonal and intra-seasonal variability in Fig. 3?

These sentences have been clarified. The term has been changed as suggested to clarify this point.

Page 6, line 29: datasets are actually four.

Page 8, line 4: "depth of wet day"?

Page 8, lines 22-24: here again you discuss TRMM performance, but discussion should be expanded.

It is obviously difficult to conclude that the radar explains all the results. But it is worth to note that active instrument provide a more accurate datasets. This is especially true for the wet events and not for the dry ones.

Page 9, line 8: sentence unclear.

We agree that this sentence were unclear and the entire paragraph has been modified.

Page 9, line 19: how do you define "extreme rainfall deficits"?

This was used as a synonym of the dry spells but we agree that can generate some confusions. The sentence has been rewritten.

Page 9, line 22: according to Table 1, not all the products are upgraded (BTW this is just regridding, so another word should be used).

We define upgridding when the regridding is done to a coarser resolution. For most of the datasets, the regridding is actually an upgrading. Nevertheless, we have clarified that some datasets are just regridded.

Page 10, line 5: as in the Abstract, please clarify this sentence.

Page 10, line 14: same sentence on TRMM performance, but no discussion.

According to the comments of the reviewer. We have modified and clarified all the sentences related to the radar of TRMM. See previous comments.

Figure 4: what is depicted here? Yearly averages?

This is the average number of dry spells (DS and DSC) per year collected on all grid points. Thus the boxplots illustrate the spatial variability of these indicators. The caption has been clarified.

Figure 5: what is displayed here? What does the y-axis refer to?

Figure 8: what do the y-axis refer to? Are monthly values represented?

Table 1: check table header.

Technical corrections

Language is below acceptable standard, and the paper requires a substantial revision in this sense, therefore I omit here to indicate individual issues.

---

## Referee Comment (RC2) · Anonymous Referee #2 · 21 Oct 2019

This manuscript compares the representation of wet and dry spells in Senegal based on rainfall estimates from different sources ranging from satellite retrievals to rain gauges and reanalysis data. Even though the manuscript initially aims to address the very important question of robustness of different metrics for wet and dry spells given the spread between available rainfall datasets, it does not succeed in delivering any coherent message. Unfortunately, the language does not allow to scrutinise the results and overall presentation is below-acceptable. My recommendation is a fundamental revision of this manuscript with hopefully some in-depth involvement of the co-authors in presenting, interpreting and structuring the results. In its current state I recommend to reject this manuscript for publication in NHESS.

[Figure]

General comments:

- The language of this manuscript is imprecise and altogether not of acceptable quality, which makes it difficult to review in the first place

- Interpretation and putting into context of the presented work / comparison with existing literature on dry and wet spells in the region is essentially entirely missing

- The goal of evaluating the performance of different datasets for dry spell and wet spell identification was not achieved as there is hardly any information that goes beyond a pure description of given plots, leaving it to the reader to come up with a conclusion

- Figures need to be revised (axis, overall readability, caption descriptions, annotations and plots cut off)

- Conclusion only summarises plots all over again

Specific comments, which are not exhaustive:

p5 Methodological approach: This section is too short – were the metrics calculated on the entire time series / per month / per pixel etc? Please also say something about the usefulness of those metrics. For example, WS99P seems to be an unnecessary metric as, per definition, the "number of extreme days" is will be 1% of the number of identified >=1mm wet days.

- Could you state the rationale behind looking at 90 - 99th centile 'wet spells' only, rather than including lower thresholds that are agriculturally relevant?

- Please show the number of rainy days per dataset as that's what the other metrics are based on

p5 ll 25-26: "The duration categories of wet spells are chosen to correspond to the different synoptic systems causing rain in West Africa (Froidurot and Diedhiou, 2017)." What does that mean? How does it address the "different synoptic systems"?

p6 ll13-14: why does the TRMM radar explain this?

p6 p15: Please indicate on the map where the Peanut Basin is

Figure2: While larger patterns are reflected in the datasets, local differences in the transition zone can be large. Please consider to show all datasets as a difference from OK (since OK is used as reference dataset later on) to help the reader spot biases more easily

p6 17-19: how was this % of dry days calculated? Is it this per day and pixel or is a "dry day" when the entirety of Senegal is < 1mm?

- Generally, please consider adding 1-2 further maps that can take into account the extreme rainfall gradient and illustrate regional differences in those metrics

p 6 l24: why is this a paradox? Also, what is the take-away message for the reader from this section?

p6 l26: " Fig. 3 also illustrates a higher variability from the seasonal scale to intra-seasonal scale." please explain more clearly

Figure 4 has boxes cut off – please replot

p7 ll11-12: "The seasonal cycle of dry spell shows slight differences between products which confirm that this eventscharacterize false start and early cessation of season in Senegal. " Please explain more clearly

p7 ll13-14: what does illustrate the severity of DSC and DS? What is this severity?

P7 l16 It's rather within the seasonal cycle than at a given date

Figure 6: what is the x axis? Where does the "0" belong? Please provide complete descriptions in your figure captions

Figure 10: why does the daily rainfall only start at about 15mm per day?

P9 l5: worth to note that WSM 99P is only the rarest because it is the only defined wet

spell metric that does not have a predefined number of occurrences per definition of percentile thresholds

p9 ll 8-9: please explain the part with the fraction more clearly

Generally, this is a section when having a map would be interesting in order to see in which region the dry spells are particularly hard to catch for certain datasets

Figure 12: It's very difficult to make out the different datasets in this plot (please improve) and the last two years of OK look rather questionable - can you comment on that? Please explain where you see a clear increase in this.

---

## Author Comment (AC1) · 13 Dec 2019

**General comments**

This paper analyses the occurrence of wet and dry spells in Senegal estimated from a set of precipitation products (in situ, satellite and reanalysis). Although the purpose of the paper is valuable and results potentially significant, the overall quality of the analysis and presentation is below acceptable standard.

We first thank the reviewer for its valuable contribution and evaluation of the document. We have modified and corrected the document as suggested to reach the standard of the journal. Significant improvements have been done and clarifications have been

included. Please find the responses of the general and specific comments in red.

1) Language needs substantial revision and improvement, by a native speaker if possible;
we have checked the document with a native english speaker.

2) The presentation of the method (description of kriging in Section 1.1 and definition of Wet and Dry spells in Section 1.3) needs improvement, the description of the methods should be expanded and the presentation clarifyed;
These subsection have been modified and additional informations have been provided.

3) The presentation of the results is often unclear, and results themselves are not discussed enough: Section 2 is basically a description of the figures, and the keypoint of the paper, i.e. the comparison of the precipitation datasets, is never actually addressed;
First, the figures have been modified. We have also modified the section 2 with more discussions on the results and less descriptions. That implies more discussions about the comparison of precipitation. Nevertheless, the reviewer mentions the keypoint of the paper is the comparison of the precipitation datasets. This is not correct. The main objective is the comparison of the detection of wet and dry spells, that are more specific that precipitation comparisons. We have clarified this point in the objective of this study.

4) Figure captions should be improved;
Modified as suggested.

5) Conclusion section is not conclusive at all, it is just a summary of the paper.
We have improved the conclusion section by providing the keys messages of this intercomparison. That results are really relevant for users and researchers working over these regions or with theses datasets and provide the uncertainties that are related to the monitoring of extreme wet and dry spells.

Interactive
comment

For the above reasons I cannot recommend the publication of the paper in NHESS. I suggest the authors to undertake a substantial revision of the paper, and resubmit it for new consideration. I list below my specific comments that I hope can help the authors in improving the paper.

We thank again the reviewer for providing these useful comments . We took into account all of them to substantially improved the quality of this study. These changes increase significantly the quality of the paper and, we hope, they allow to reach the standard of the journal.

**Specific comments**

Page 1, line 6: "more variability", do you refer to some intra-dataset variability? If this is the case, this expression is not suitable, please rephrase.

Indeed, the term "more variability" is used to illustrate the differences between products.The sentence has been rephrased to clarify.

Page 1, lines 9-10: these lines are unclear.

All the paragraph has been rewritten to correct an to clarify the text.

Page 1, line 15: add the reference to Taylor et al. 2017, https://www.nature.com/articles/nature22069.

Modified as suggested

Page 1, lines 15-16: this sentence should be moved at line 19, before "Recently extreme events: : :"

Modified as suggested

Page 2, lines 25-27: sentence unclear.

According to the reviewer comments, all the paragraph has been rewritten.

Page 2, line 30: when stating the objective of the paper, it is not clear that you compare satellite/reanalysis products to rain gauges, which you consider as reference. Please clarify this point.

[Figure]

We thank the reviewer for this comment. Indeed, this important point need to be clarified. This paper aims to make an inter-comparison between several products from observations, satellite data and models. More specifically, the objective is to compare them on their ability to detect potentially high-impact dry and wet events occurring in Senegal. However, according to literature and the data characteristics, there is no ideal product, especially when studying strong to extreme events. Indeed, the network of surface observations is too coarse, the products derived from satellite datasets are sometimes unprecise either because of approximations (due to some proxy used) or because of orbital characteristics (revisit period, resolutions), and finally the models are well known to have uncertainties, especially over the tropical areas. Thus, it seems difficult to say which product is the reference even if ground observations are expected to be an ideal candidate. However, the uncertainties associated with krigging technics, highlighted in our study, show that even these products should be used with caution.

Page 2, line 33: are wet and dry spells "extreme hazards"?
Dry and wet spells are defined with different intensities and durations. Thus, events can be considered moderate to extreme for the highest intensities and/or the longest durations. This work is a first step in identifying potential high impact events. It is therefore important of having a large sample of different events. However, most of the results presented in the study focus on events that have a return period of about ten years. We have clarified this point in the text.

Page 3, line 1: DS, DSC, WS and WSC are not defined in the text.
Sorry for this mistake. Every wet and dry spells is now well defined in the text.

Section 1.1: do rain-gauge time series have passed any objective homogeneity check?
The rain-gauges used in this study have passed two levels of quality control. The first one is a manual check for suspicious records was carried out and then additional checks were carried out, including verification of station locations, identification of repeated data, identification of outliers, comparative tests using neighbouring stations and search for suspicious zero values (missing data or zero precipitation). This information has been added in the text.

Section 1.1: BK is poorly described.
A better description of the technical procedure to generate block krigging is now added. This description gives sufficient information for the reader to understand the technic. The details of this technic are described in supported litteratures cited in the new version.

how the lambda weight are assigned?
The basis of kriging is the variogram which is a function of the variance and distances between stations. This function gives two properties, the nugget effect which corresponds to the limit of the variogram in zero. The nugget effect represents the variation between two very close measurements and the range that corresponds to reaching a plateau indicating that there is no longer any spatial dependence between the data. Depending on these two parameters we have the screen effect: the nearest points receive the most important weights (lambda). This screen effect varies according to the configuration and the variogram model used for kriging. The greater the nugget effect, the less screen effect there is.

Page 3, line 18: why kriging reduces high values and increase low values? Please explain. Equation 2: what is xq?
Kriging tends to greatly smooth intensities. This effect is well known and tends to reduce extreme wet and dry events and converge to the mean values, especially when the network of rain-gauges is coarse. In consequences, the intensities are reduced as well as the occurrence of zero rain. The effect is similar to a smoothing when applying mean values. Equation 2 is a bias correction using the quantile mapping method. This method corrects the underestimation of high rainfall intensities by kriging. xq are the kriged data values; xo are the observed data values. The purpose of quantile mapping is to bring the two Cumulative Distribution Function (CDF) together. By correcting the distribution of the kriging outputs, percentiles and high intensities are corrected as well.

Page 3, line 22: square root of OK variance is used for what?
Square root of variance is a derived product for the kriging technic. It is also commonly called kriging error. It was used with a threshold of 0.5 to identify regions with low rain gauge density. Looking at the results in Figure 2, it can be seen that the eastern central part of the country is ignored because the error on kriging is less than 0.5. This choice is essential for the reliability of the data collected.

Section 1.2: there are discrepancies in dataset resolutions in the text and Table 1, please check.
Corrected

Page 5, line 9: where did you define wet and dry spells as extremes? "A maximum number of definition", this sentence is unclear.
We agree that these sentences were not clear. The entire paragraph has been rewritten.

Page 5, line 16: on which statistical basis do you define "DS extreme long"? You should use the "extreme" word carefully, and only after an analysis of distributions.
We agree with the reviewer, and the term "extreme" is now used with more caution. Nevertheless, it is worth noting that the selection of dry and wet spells is based on the PDF of the precipitation from 1991 to 2010 to select some of the most extreme events. It is, obviously, depending to the criterias but most of the events have a 10-y return period.

Page 5, line 20: what do you mean by "DSC duration is known"? Please clarify.
We have clarified that DSC have fixed durations, from 5 to 20 days.

Page 6, line 12: why BK is more comparable to averaged values and more in agreement with satellites?
According to the literature (Lloyd and Atkinson, 2001; Maidment et al., 2013;panthou et al., 2018), the two methods of kriging are different: Ordinary Kriging (OK) estimates rain as a punctual value while Block Kriging (BK) estimates rain on space blocks with

an average value rainfall. Rainfall satellite estimates are closer to an average value over a space block than a punctual value. For this reason, the BK should be closer to satellite datasets.

Page 6, line 14: please explain why radar on board of TRMM explain TRMM performance.
The TRMM satellite is the first satellite with an active radar instrument onboard. It is a powerful added value since it provides a profil of the rainfall activity. This is especially important over tropical region where unsateurated downdraft and evaporation of the rainfall is important. It is also important since the rainfall estimates is not based on a proxy of the top of the convective cells. This estimation is quite common and derived from passive radiometer but shows some bias (overestimation) during the collapsing period of the convective cells. This point is now clarified in the document.

Page 6, line 15: please improve the description of the Peanut Basin or highlight it on the map, to facilitate the reader to locate the region.
Modified as suggested

Page 6, lines 20-21: This actually means that during dry season date are not collected, isn't it? In this case you cannot consider these days as dry.
This is correct. We have clarified the document as suggested.

Page 6, line 26: does "variability" refer to the datasets? I suggest to find another expression in this case. How can I compare seasonal and intra-seasonal variability in Fig. 3?
These sentences have been clarified. The term has been changed as suggested to clarify this point.

Page 6, line 29: datasets are actually four.
Corrected as suggested

Page 8, line 4: "depth of wet day"?

Modified by "rain amount of day"

Page 8, lines 22-24: here again you discuss TRMM performance, but discussion should be expanded.
It is obviously difficult to conclude that the radar explains all the results. But it is worth to note that active instrument provide a more accurate datasets. This is especially true for the wet events and not for the dry ones.

Page 9, line 8: sentence unclear.
We agree that this sentence were unclear and the entire paragraph has been modified.

Page 9, line 19: how do you define "extreme rainfall deficits"?
This was used as a synonym of the dry spells but we agree that can generate some confusions. The sentence has been rewritten.

Page 9, line 22: according to Table 1, not all the products are upgraded (BTW this is just regridding, so another word should be used).
We define upgridding when the regridding is done to a coarser resolution. For most of the datasets, the regridding is actually an upgrading. Nevertheless, we have clarified that some datasets are just regridded.

Page 10, line 5: as in the Abstract, please clarify this sentence.
We agree that these sentences were not clear. The sentence has been modified.

Page 10, line 14: same sentence on TRMM performance, but no discussion.
According to the comments of the reviewer. We have modified and clarified all the sentences related to the radar of TRMM. See previous comments.

Figure 4: what is depicted here? Yearly averages?
This is the average number of dry spells (DS and DSC) per year collected on all grid points. Thus the boxplots illustrate the spatial variability of these indicators. The caption has been clarified.

Figure 5: what is displayed here? What does the y-axis refer to?

This figure illustrates the seasonal cycle of occurrences of dry spells (DS and DSC) from 1998 to 2010 over all grid points. This occurrence was previously indicated in term of number (depending the number of years and grid cells used). In the new version of the document, the frequency of the event, in relation to the total number of event, has been provided. This indicator, in percent, is more understandable.

Figure 8: what do the y-axis refer to? Are monthly values represented?
Same answer as with Figure 5. Again, the new figure and caption, on the frequencies of these events, facilitate the comprehension of these results.

Table 1: check table header.
Modified as suggested

Technical corrections
Language is below acceptable standard, and the paper requires a substantial revision in this sense, therefore I omit here to indicate individual issues.
The entire document has been revised and clarified. A native English speaker has corrected the English.

---

## Author Comment (AC2) · 13 Dec 2019

**General comments**

- The language of this manuscript is imprecise and altogether not of acceptable quality, which makes it difficult to review in the first place
We understand the reviewer's opinion. To correct these errors we had the entire article proofread by a native English speaker. We have thus made significant changes to the text to ensure that it is in line with the journal's requests.

- Interpretation and putting into context of the presented work / comparison with existing

literature on dry and wet spells in the region is essentially entirely missing

We partially agree with the reviewer. Significant efforts have been made to better contextualize this work in relation to previous work. However, the bibliography on this topic in this region is very limited and we have focused on the articles that are relevant to this study. Through the review, no specific study is proposed. We hope that the integrated articles will be satisfactory for reviewer.

- The goal of evaluating the performance of different datasets for dry spell and wet spell identification was not achieved as there is hardly any information that goes beyond a pure description of given plots, leaving it to the reader to come up with a conclusion

Here again, we partially agree with the reviewer. Indications on the quality of detection of wet and dry spell have been made in this study and, in the conclusions, recommendations are made. However, we agree with both reviewers that it is necessary to do a better synthesis work to avoid too much descriptions and to discuss in more detail the conclusions and origins for these results. This is why a large effort has been made in section 3 (results, simplified in the new version) and section 4 (discussion, where the consequences and possible reasons for these results are discussed). The conclusion section has also been modified to highlight the most significant results.

- Figures need to be revised (axis, overall readability, caption descriptions, annotations and plots cut off)

All figures have undergone quality control to meet the requirements of the journal.

- Conclusion only summarises plots all over again

As previously mentioned, the conclusions has been entirely rewrote to focus on the key results of this study.

**Specific comments, which are not exhaustive:**

p5 Methodological approach: This section is too short – were the metrics calculated on the entire time series / per month / per pixel etc? Please also say something about the
usefulness of those metrics. For example, WS99P seems to be an unnecessary metric as, per definition, the "number of extreme days" is will be 1identified >=1mm wet days. We agree, this section needs to be developed, particularly on the indicators developed. Indeed, all DS and WS indicators are calculated for each pixel from the daily data. The usefulness and the objectives of having a large spectre of indices (from mild to the most extreme definitions) is now better justified. WS99Ps are the most extreme events we could detect in our analysis. Because of their potentially extreme impact we have to include them in our analysis even if they are rare.

- Could you state the rationale behind looking at 90 - 99th centile 'wet spells' only, rather than including lower thresholds that are agriculturally relevant?
That is a relevant question. The purpose of this paper is to provide a wide range of potentially high-impact event indicators. Thus, we focused on potentially impacting wet (WS) and dry (DS) events. DS are studied because their presence and duration can generate a rain deficit and droughts that can impact yields. WSs are able to destroy seedlings and crops through heavy rainfall or subsequent flooding. The detection of DS is based on previous studies that define these periods as periods of non-precipitation.

- Please show the number of rainy days per dataset as that's what the other metrics are based on
This information is already provided by the distribution of dry days (Figure 3) and the distribution of cumulative rainy days (Figure 10). The text has been revised to clarify this point.

p5 ll 25-26: "The duration categories of wet spells are chosen to correspond to the different synoptic systems causing rain in West Africa (Froidurot and Diedhiou, 2017)." What does that mean? How does it address the "different synoptic systems"?
In this sentence, we justify the durations of the wet spells according to previous studies that highlighted synoptic origins of the rainfall. One of the most important drivers of the mesoscale convective systems is the African Easterly Jet. This perturbation generate a signal at around 3 to 5-d period. Nevertheless. We have completely modified this

paragraph to clarify this point.

p6 ll13-14: why does the TRMM radar explain this?
The TRMM satellite is the first satellite with an active radar instrument onboard. It is a powerful added value since it provides a profil of the rainfall activity. This is especially important over tropical region where unsateurated downdraft and evaporation of the rainfall is important. It is also important since the rainfall estimates is not based on a proxy of the top of the convective cells. This estimation is quite common and derived from passive radiometer but shows some bias (overestimation) during the collapsing period of the convective cells. This point is now clarified in the document.

p6 p15: Please indicate on the map where the Peanut Basin is
The basin is now indicated in the map as suggested

Figure2: While larger patterns are reflected in the datasets, local differences in the transition zone can be large. Please consider to show all datasets as a difference from OK (since OK is used as reference dataset later on) to help the reader spot biases more easily
Modified as suggested

p6 17-19: how was this "dry day" when the entirety of Senegal is < 1mm?
Thank you for the question, this percentage of dry days is the number of dry days compared to the total number in each month averaged from 1998 to 2010 and on all pixels in Senegal. Note that a dry day is defined as any rain <=1 mm. This has been clarified in the caption of the figure and in the text.

- Generally, please consider adding 1-2 further maps that can take into account the extreme rainfall gradient and illustrate regional differences in those metrics
As suggested by the reviewer, additional figures have been added in the supplementary material to address this point. Nevertheless all types of uncertainties and case cannot be discussed and plotted in the main document.

p 6 l24: why is this a paradox? Also, what is the take-away message for the reader from this section?

The paradox is illustrated by the fact that the products with the lowest number of dry days in our study (namely TAMSAT and CHIRPS) tend to underestimate seasonal accumulation. This is particularly true in the South. The take away message, related to the uncertainties of the detection of wet and dry spells, and about the quality of each product, is clearly provided in conclusion.

p6 l26: " Fig. 3 also illustrates a higher variability from the seasonal scale to intraseasonal scale." please explain more clearly

The idea is to show that despite a fairly good agreement on the spatial distribution of seasonal accumulation, the intra-seasonal distribution of dry days shows larger disparities between products. The sentences have been rewritten to clarify this point.

Figure 4 has boxes cut off – please replot

Corrected as suggested

p7 ll11-12: "The seasonal cycle of dry spell shows slight differences between products which confirm that this events characterize false start and early cessation of season in Senegal. " Please explain more clearly

In this sentence, we relate the detection and the anomaly of dry spells and some other characteristics of the rainy season, namely the false start (when a dry spells occurs just after the first rainfall event of the season) and cessation. As mentioned previously, this paragraph has been rewritten to clarify this point.

p7 ll13-14: what does illustrate the severity of DSC and DS? What is this severity?

These rainy breaks (dry spells) can be detrimental to yields, especially when they occur at the beginning of the season, causing farmers to lose seedlings. In addition, they can occur in the middle of the crop maturation season to generate water stress by lowering the WS (Water satisfaction) of the plants. The severity of dry spells defines their potential to be dangerous. Due to the nature of the precipitation values, it is not

possible, unlikely wet spells, to assess this severity by using the intensity of the rainfall during dry spells. So they are only estimated by the duration of events. In this study, we have selected the longest and therefore the most severe DSC and DS.

P7 l16 It's rather within the seasonal cycle than at a given date
Corrected

Figure 6: what is the x axis? Where does the "0" belong? Please provide complete descriptions in your figure captions
Figure 6 is a typical Taylor diagram. Radius expresses the standard deviation, the angle the correlation and the distance from the bottom right point, the RMSD. The caption has been clarified as suggested.

Figure 10: why does the daily rainfall only start at about 15mm per day?
Sorry for the missing information. We started the analysis with precipitation above 10mm. This is now mentioned in the text and in the caption.

P9 l5: worth to note that WSM 99P is only the rarest because it is the only defined wet spell metric that does not have a predefined number of occurrences per definition of percentile thresholds
This is correct. We have mentioned that in the modified version.

p9 ll 8-9: please explain the part with the fraction more clearly
The fraction related to the total number of data available has been clarified. Due to the comment of the first reviewer, this paragraph has been deeply modified.

Generally, this is a section when having a map would be interesting in order to see in which region the dry spells are particularly hard to catch for certain datasets
We thank the reviewer to point out this good remark. This map has been added

Figure 12: It's very difficult to make out the different datasets in this plot (please improve)
We agree there is a lot of information in that figure. Nevertheless, we have modified it.

[Figure]

and the last two years of OK look rather questionable - can you comment on that? Please explain where you see a clear increase in this.
As suggested, we have discussed the issues of the recent years using OK that could be due to the missing datasets.

---

## Referee Report (RR1)

This manuscript greatly improved after the first round of revision and now illustrates the capability of different rainfall datasets to represent high-impact dry and wet spells rather nicely. Particularly the discussion of good and bad performing datasets in view of their data sources is very informative and this paper importantly illustrates the difficulties in deriving information on drought and rainfall extremes, even from the wealth of rainfall datasets available nowadays. It is therefore an important addition to literature, particularly for West Africa where analyses of extreme weather metrics from available rainfall observations become more and more important but reliability of datasets for such applications is too rarely questioned.

Unfortunately and in spite of the authors stating that the manuscript was properly proof-read and even checked by a native speaker, non-cosmetic corrections are still necessary in high density. I would urge the authors to read the entirety of this manuscript again – to actually do it and to add all the –s and cross the ts. It is rather tedious for a reviewer to do this job and was very much not enjoyable. We all know that if only half of those errors make it through to publication it reduces the trustworthiness and readability (and therefore the impact) of this study, which is completely avoidable.

I have a couple of major comments regarding the dry spell results that need to be clarified and the minor comments need to be addressed before I can recommend this manuscript for publication.

Major comments:

- I have methodological concerns (1) regarding the definition of dry spells causing large spreads at the end of the year, inconsistent with early in the year (Fig5) and (2) whether the inclusion of the dry season into metrics skews the aggregated results for DSC10, DSC20 and DSxl (Fig 4,6,7) away from anything that is relevant regarding "drought hazard" or "high impact event" if they are dominated by the dry season signal. Please clarify the behaviour of (1) and illustrate that the dry season sensitivity is not dominating Fig 4,6,7 results (I.e. I'd like to see Fig4,6, for 'rainy' months only, say April-Nov or similar).

General comments:

- Different from the author's answer, there were in fact no additional maps of any metrics added to the supplementary material that would illustrate the spatial pattern of discussed metrics. Could you please add those as mentioned in the response?
- Please add letters to figures and refer to panels in plots consistently throughout the manuscript.
- Where possible provide some key quantification of discussed discrepancies and biases. Currently the analysis remains predominantly qualitative.

Minor comments (those corrections are unfortunately not exhaustive!):

P1 line 6 remove first "same"

P1 line 10 "while," -> meanwhile

P1, Line 10: why does this suggest an "early cessation" of precipitation – what would be a normal cessation? Doesn't this rather say that rainfall is more intermittent during the cessation of precipitation ("false onset" expression captures that)

P1, LL 10-11: remove "while," , the strongest contrasts between the data products [..] observed _for_ the amplitude

P1, Ll 12-13: remove "quite similar", better to say "show a comparable/similar frequency of wet sequences"

P1, L 18 thE Sahel

P2, l 7: "will increase significantly [..] during the lean season of 2018" what does this mean? Is it projected to increase in the future or did the number increase in 2018?

P2 ll14-15: "these hybrid rainy seasons**,** illustrating a rainfall regime intensification **, ARE** part of"

P2 l34 remove "although", it's "however"

P3, l7; l28; p4 l3: techniques (same for all other cases of "technic")

P4 l3: remove "is" from "is uses a moving neighborhood"

P4 l5: centered **AT** the block mean

P4 l7: "has a minimum variance of estimation error" - does that mean it minimises the variance of the estimation error? In any case, please clarify

P4 ll 9-10: This should be changed to "we use the kriging RMSE to mask out areas [..]" etc. The part with "can be possibly masked out" is not fitting since Fig1 indeed shows vast masked out areas, so it is applied rather than "can potentially be applied"

P 4, l32: resampled to 0.25 (remove < ), same p5, line 1

P5 l16: what kind of sensitivity is meant here and how is the analysis of those datasets related to future impacts?

P5, l19 use**d**

P5, l20 precipitation IS lower

P5, l23 descripted -> shown

P5, l23-24 during a specific period(**s**) - remove s - and **IS** called

P5 26 when the rainfall IS not sufficient … and therefore DOES not provide..

P5, l28 The results presented in THIS study

P5, l29 from the other durationS

P6, l5 "and because of the synoptic systems associated.." this should be better explained and the characteristics of MCSs (e.g. propagating and therefore extremes rarely stationary etc) mentioned. It's otherwise not clear for people who don't work in the region

P6, l10-11 "these periods are defined according to the different synoptic components that drive the rainfall variability" again, this is very vague. If there is related reasoning it should be stated explicitly, and those factors at least mentioned (preferably with a reference about the importance of that factor)

P6. L21: remove "the" from a south-north gradient

P6 l22 in term**S** of

P6 l22-23 It would be good to state in the text that this is rainfall values for June-October, possibly right in the introductory sentence of the "seasonal rainfall" section

P6 l23 closeD -> remove D

P6 l24 kriged observed precipitation **datasets** (better: in-situ datasets or rain gauge datasets etc)

P6 l25-26 Our results from CMORPH (..) - there are several language problems in this sentence, please correct (confirmS, "which" showed, "these" precipitation)

What is the result here for CMORPH? I assume this refers to lower seasonal precipitation compared to the gauge-based products but it's not stated. It would be useful to quantify "the results are close" or "underestimating" by giving a percentage range for the rainfall differences between those datasets, or correspondence in pattern correlation or anything that underpins the qualitative statements in this section.

P6 l32-33 "When looking over smaller areas differences are more important and any of the products is able to get this structure even if their bias stay low" Please correct (language problem) and clarify this sentence, which region this refers to and where biases stay low. I'd assume this means something like "Regional-scale patterns in rainfall are of particular importance. All products seem to approximately agree on the magnitude of spatial rainfall variation. Such variation is particularly pronounced across the peanut basin, for which the bias between rainfall products is low" - again, can "low" be quantified? It seems difficult to assess those statements by just eyeballing the maps and no indication of what the authors refer to.

P7, l8 closest to BK in intensity - can "closest" be quantified, just to give the reader some idea what the magnitudes here are in terms of biases, agreement etc.

P7, l14 Does this paragraph now refer to Fig3? Reference missing

P7, l19 accurate productS – remove S

P7 20-21 On P6, 25-26 it says that CMORPH misses local convective rainfall between scans, resulting in somewhat lower seasonal total rain, and here it says it tends to overestimate small (low-intensity?) precipitation, where the authors say "which would explain why the difference appears here but not when looking at the cumulated rainfall". This can be confusing and would

be worth clarifying. I think this says that high-intensity rain dominates the wet season, of which CMORPH misses events in-between scans, but low-intensity events during the dry season are overestimated. But please state this more clearly.

P7 l23: in termS of, better would simply be "This is also visible for the cumulated rainfall"

P7 l27 finalLy

P7 l29: it is a difficult – remove "a", better than "to find the reasons" would be "to suggest an explanation for"

P8 l7 different typeS, depending on their

P8 l8-10 "In the main document"..  and reference to supplementary can be shortened to "We focus on.." with (see Table 2 for the definitions, further results in supplementary material). "Nevertheless [..]" can be dropped.

P8 l13: dry days is Fig 3, not Fig4

P8 l13 "This is in agreement with the previous result" - as the authors show later on, this is not in agreement regarding the CMORPH / TRMM behaviour, which so not agree for the dry days.

Fig 4 caption: "Boxplots of the average number [..] the left and right edges of the box" this should be bottom and top edges. What does "extreme values" for the whisker position mean. Min and max? Is this really per year or again from June-October like Fig2? If it is per year, wouldn't the dry season performance shown in Fig2 predominantly affect those extreme dry spell indices?

P8 l16 "than TAMSAT and CHIRPS" replace with "as"

P8 l19 cloud top temperature

P8 l 20 MO was already introduced in l15

P8 ll 21-22 This can explained ...compared to the observations -> This may explain the relative good performance [..]  compared to the gauge observations

Fig5: how is this frequency defined? Description in caption and text just says "seasonal cycle of dry spells" without further specification. Also, why are there such inconsistencies moving from Dec to Jan? Particularly visible for DCS10, 20 and DSxl. Is there in problem in how the dry spells are identified at the end of the year? Must be a methodological issue that the spread is large in Dec and gone in Jan. Does this affect the aggregated metrics in the other plots?

P8 l25: It is a very important point that those dry spell metrics are so strongly affected by the dry season and should be pointed out much earlier in the manuscript. While the behaviour of the datasets during the dry season is interesting (and sufficiently shown in the seasonal cycle plots), the importance of dry spells depends on whether they appear during the wet or dry season. For example, Fig4 shows that DSC20 is around 1 or below per year, questioning the usefulness of this metric in the hazard context. It suggests that this metric reaches "1 occurrence per year",

which likely reflects the dry season - this is not very interesting and not reflecting an "extreme event". On the other hand, it would be an important information if this event occurred once a year during the monsoon season. How much are the dry spell results skewed towards rainfall dataset dry season skill (affected by low-intensity precipitation breaks rather than MCSs)? Why weren't the non-seasonal cycle plots restricted to June-Oct (or at least months outside the dry season)?

P8, l31 in agreement FOR the observations

P8 31-34 "The evolution of DSl **is also interesting by focusing** on relative **mild droughts with specific durations that are sensitive to dry spells** during the onset and retreat phases of the monsoon. This detection is, **by far, the more variable** from one product to another. For this **specific drought** it is difficult to **distinguish specific behavior** of a group of products. Each possesses a **specific time evolution** [..]"

**Please improve wording.** [..] is also interesting as it represents/characterises relatively mild droughts with a fixed duration. This metric is most sensitive to dry spells during [..], and is by far the most variable [..]. For this dry spell metric, it is difficult to distinguish any specific behavior [..]. Each possesses an individual time evolution [..]

P9 ll3-5: **gauge** observations (the difference to satellite observations is otherwise not clear). Indeed, the difference between the interpolated gauges is remarkable and, if ignoring ERA5, almost as large as the spread between the satellite observations. Again, it would be worth to quantify this uncertainty in the text. Looking at DSl at the hight at the rainy season between Aug-Sep, the frequency difference between BK and OK is around 20%. The dry day frequency increases by more than 100% just changing from BK to OK, based on the same set of stations. Please be more explicit in numbers about statements rather than to rely on handwaving only

P9 l10: spatial datasets -> gridded datasets

P9 l11: are providing in -> are provided in THE

Fig6 caption: it should be BK which is mentioned as reference dataset here. What is the x-axis? If it is standard deviation too the ticks should be similar to the y-axis.

Is it correct that this diagram was calculated from the spatial maps (like Fig2) of those metrics and e.g. spatially correlated? Which leads me to the question why no metric map was added to the supplementaries (contrary to what was stated in the reviewer response)?

Again this relatively good aggregated agreement may be artificially boosted by including the long dry season. What would this look like for the rainy period only (or say April-Nov?). I think it doesn't reflect well what was shown based on the seasonal plots and distracts from the fact that discrepancies are large when it's most important.

P9 l12-13 "For the DSC10 and DSC20 and the DSl there is no clear difference amongst the datasets. However, DSC10 is more sensitive to the datasets."

Is this supposed to refer to DSC20, DSl and DSxl, which all sit in the area of low standard deviation? The spatial correlation for those metrics seems rather low compared to DSC10

P9, l22 similitude -> similarity

P9 ll23-24 "Finally, DSl displays a specific time evolution." -> displays a time evolution that seems distinct from the other metrics?

P9 l32 observations -> in-situ / gauge observations

P10 l3  I would suggest the authors add lettering to their plots and refer to Figx a,b etc throughout the manuscript. That would make it much easier to follow which panel is being discussed without having to check and recheck the acronyms.

P10 l7 I think that should read "WS1 99P" instead of WSl

P10 l10 "This distribution shows to see tipping points on daily rainfall." language problem, please rephrase

P10, l21 except to the -> except FOR the WSM

P10 ll24-25 contributes bias correction -> allows for such biases to be taken into account

P11 l3 in-situ observations

P11 l4-5 are more likely to be compared with -> are more likely to be comparable to gridded [..]

P11 l25 the monitoring [..] are compared -> the monitoring [..] is tested  OR the representation [..] is compared

P11 l26 3 products BASED on raingauges

P11 l27 by upgrading or -> by area averaging, interpolation or[..]

P11 l29 THE large-scale climatology

P11 l33 for an average rainfall like most of -> remove "an", like FOR most of

L33 this good agreement start to dissipate -> startS

P12, l2 "It turned out that each of the kriging methods were positioned in these groups." -> Interestingly, from the kriging methods each falls into one of these groups.

P12 l10 "However, there is less agreement between the different data products for dry spells than for the wet spells." Shouldn't this be "there is MORE agreement for dry spells than for wet spells" ?

P12 l14 "record the rainiest days but minimize these high rainfall events."  -> record highest rainfall intensities but show lowest rainfall frequencies (?), otherwise please clarify what that means.

---

## Author Response (AR2)

**Author's response to anonymous referee 1**

**General comments**

This paper analyses the occurrence of wet and dry spells in Senegal estimated from a set of precipitation products (in situ, satellite and reanalysis), aiming at assessing the performance of the analysed datasets.

This is my second review of the paper and, although I find it slightly improved compared to the first submission, I think that the overall quality of the analysis and presentation is still below an acceptable standard. The revision process has been particularly difficult due to inappropriate and sometimes confusing wordings and lack of explanations. I recommend the authors (in particular the senior scientists in the team) to take care of the quality of the manuscript for next submissions, to make the reviewer's work easier and more effective. I highlight the importance of taking seriously this point, because lack of clarity makes more difficult (sometimes impossible) to evaluate the goodness of your results and the relevance of your findings.

Although the purpose of the study is valuable and results potentially significant, I believe that the paper still needs major and substantial revision before to be published.

First of all, we thank the reviewer for his valuable contribution and his evaluation of the document. We have deeply modified and corrected the document as suggested to meet the standard of the journal. Significant improvements were made and clarifications were included. The explanation of the results is more orderly to improve their consistency. We checked the document with a native English speaker again. Please find the responses to general and specific comments in red.

**Specific comments**

In general, English language still needs fixings, many sentences need clarification, method and analysis need further explanations, and unnecessary repetitions should be eliminated.

We apologize for the mistakes related to the english. The text has been corrected by a english native speaker.

Throughout the text, significance word is often used, whereas statistical significance of the results is never assessed, weakening the credibility of the conclusions.

"Significant" will be used when we perform significance tests only. We have decided to add two figures to the main document whereas statistical significance of the results is assessed by the correlations and Root Mean Square Error (RMSE).

The objective of the paper needs to be clearly declared. Specifically, while reading the manuscript, it is not clear whether observations are taken as reference. This is clarified in the Discussion Section, when it is stated that BK is considered as reference. This choice must be justified and clarified at the beginning of the manuscript.

We thank the reviewer for this comment. The objective of the paper is clarified as follows:
"This paper aims to set up an inter-comparison between several products resulting from observations, satellite data and models. More specifically, the idea is to compare their ability to detect potentially high-impact dry/wet events in Senegal. The reference dataset chosen for this study is block kriging (BK), despite the imprecision of the network of surface observations and uncertainties related to kriging techniques. Although OK produces the best point-based estimates, in cases where nugget variance is great, interpolated surfaces may be subject to local discontinuities, consequently troubling longer range spatial variations. BK circumvents this by computing averaged estimates over areas or volumes (albeit at the cost of reduced spatial resolution). BK estimates may also be

more realistic since data from one point usually represents the area around it.We therefore suggest BK as the best available reference candidate."

There are large differences between OK and BK of the same dataset (see e.g. Fig. 4 and Fig. 8), this needs to be discussed.
According to this comment, this paragraph has been added in the discussion:
"Disparities between OK and BK, even though they come from the same rain gauge networks, are akin to the differences between the two kriging methods. Indeed, the OK method is used to estimate a value at a point in a region, applying data close to the estimation point, while the BK method uses a movable zone or block. Therefore, the smoothing effect resulting from kriging is stronger in BK than OK, since it tends to diminish rain event intensity and augment rainy day occurrences."

Discussion and Conclusion Sections look as a summary of the results, with very succinct discussion. We understand that TRMM is generally better than other products. However, is this true for all the metrics? Which are the datasets outperforming on specific metrics? This needs to be clarified. Moreover, implications of the results needs to be highlighted. For instance, what do users of precipitation products in Senegal learn from your results?
We agree with adding discussion about that. We have modified these paragraphs in the conclusion as follows:

"However, there is less agreement between the different data products for dry spells than for the wet spells. Indeed, the scores are more distant from the reference, BK. TRMM is particularly accurate and close to observations, and the intensity of wet events in TRMM and BK can sometimes be more than double that of TAMSAT and CHIRPS, which underestimate the intensity of these events. CMORPH and reanalyses provide fairly moderate intensities. The trend that has emerged is that satellite products that combine raingauges record the rainiest days but minimize these high rainfall events. In addition, the proximity between TRMM and BK can be explained by the radar on board the TRMM satellite. Because even though CMORPH combines infrared and microwave sensors like TRMM, it is not as close to OK or BK as TRMM. Thus, only the radar seems to favour TRMM's performance.  Finally, the climatologic trends and inter-annual evolutions are tested. There is a slight trend towards a decrease of the DS for the products and a positive but not significant trend of the WS. The too short duration for all the products that are available may explain this insignificance. This study shows that despite the general agreement on seasonal precipitation, there is a large uncertainty associated with the monitoring of extreme wet and dry spells at the intra-seasonal time scale. This study allows validating the most robust datasets, TRMM, for Senegal that could potentially be extrapolated to the whole of West Africa. This is crucial for monitoring, forecasting and determining the potential socio-economic impact of these periods of extreme drought and humidity."

have been replaced by:

"Nevertheless on WSC, TRMM maintains its affinity to OK and BK, unlike CMORPH which tends to be closer to ERA5 and NCEP. The TRMM on-board radar appears to play an important role because of its close affinity to rain gauges, especially in WS and WSC. Moreover, the WS intensities in TRMM, OK and BK are often more than double those of TAMSAT and CHIRPS. This exemplifies the difficulties of satellite datasets which use only infrared sensors. The reason for this is that cold but non-precipitating cirrus clouds impact the infrared with very cold temperatures, so the system sees these clouds as precipitating. Finally, interannual progressions of dry/wet spells were compared. We noted a slight trend toward DS decrease for the products as well as a positive but non-significantWS trend. This insignificance may be explained by the extremely short durations of the products available. This study shows that despite the general correlation on seasonal precipitation, there is extensive uncertainty about monitoring extreme dry/wet spells on an intra-seasonal timescale. Nevertheless, since there is a marked proximity between TRMM and rain gauges for all dry/wet spell categories, TRMM may be a prime candidate for extrapolating these results to other areas of West Africa. Our study reveals several potentially important implications, in particular concerning the judicious choice of
datasets to implement early warning systems (EWS) integrating a multi-hazard approach and disaster risk management plus adaptation to a "hydroclimatic intensity" context. This study also provides useful information for different hydrological and agronomic applications by defining a wide range of rainfall metrics. This may benefit agricultural insurance companies as well as stakeholders by implementing more effective indicators for considerably-improved mitigation measures."

Title: observations should also be mentioned, given that depending on the kriging method results are different. I suggest: "Wet and dry spells in Senegal: Evaluation of satellite-based, reanalysis and in-situ estimates".
We have modified the title as follows:
"Wet and dry spells in Senegal: Evaluation of satellite-based and model re-analysis rainfall estimates"

has been replaced by:

"Wet and dry spells in Senegal: Comparison of detection based on satellite products, reanalysis and in-situ estimates"

Abstract: A sentence on the implications of your results is needed.
We agree with adding discussion about that. This sentence has been added at the end of the abstract:
"Our findings provide guidance in choosing the most suitable datasets for implementing
 early warning systems (EWS) using a multi-risk approach and integrating effective dry/wet spell indicators for monitoring and detection of extreme events."

L10: How the fact that dry spells are more frequent at the beginning and end of the rainy season indicates false start and early cessation? Please clarify.
We have modified the sentence as follows:
"All datasets show that dry spells are more frequent at the beginning and end of the rainy season, indicating a false start and early cessation of precipitation." has been modified by:
"All datasets show that dry spells appear to be more frequent at the start and end of rainy seasons. Thus, dry spell occurrences have a major influence on the duration of the rainy season, in particular through 'false onset' or 'early cessation' of seasons."

L11: "wet strong rainfall events" is misleading. Do you work on wet sequences or sequences of very wet events? Please clarify.
We have modified the sentence as follows:
"While, the strongest contrasts between the data products are observed on the amplitude of wet sequences." has been modified by:
"The amplitude of wet spells shows greatest variation between datasets."

L15: "hydrological climate": I cannot find this expression in Giorgi et al. 2011. What do you mean exactly?
We have modified the term as follows:
"hydrological climate" has been replaced by:

"Hydroclimatic Intensity"

L9: "potentially high impact indicators": what do you mean exactly?
We have modified the sentence as follows:
"Thereafter, potentially high-impact indicators of dry and wet spells are defined and characterized."
has been modified by:
"In this paper, potentially high impact indicators are defined and characterized. Here we use the term "potentially high impact indicators" to illustrate the extreme dry/wet spells subject to this analysis. This term is used to better encompass vulnerability, exposure of populations and risks of hazard."

L16: How many stations are used?
We have modified the sentence as follows:
"Daily rainfall data are provided by the National Meteorological Service of Senegal (ANACIM)."
has been modified by:
"Daily rainfall data were provided by the National Meteorological Service of Senegal (ANACIM) for 65 locations covering the period 1991-2010 (Fig. 1)."

L25: "generally aggregated": This is vague. Do you mean that satellite data have finite spatial resolution?
This term "generally aggregated" has been removed in the sentence.

L29: Is the variogram of the region known? Where can we see it?
The variogram of the region has been calculated and added in the supplementary material.

L2-3: This sentence is unclear, please explain how lambda coefficients are derived.
We have modified the sentence as follows:
"Note that, the kriging weights lambda are derived from an optimization scheme containing (n+1) simultaneous linear equations." has been modified by:
"The lambda_i kriging weightings are obtained by configuring an optimization scheme containing n+1 simultaneous linear equations. These equations are derived from the standard variogram models for the distance separating sampling points from target locations using the Lagrange multiplier."

L19: "Very indirect"? Give a measure of indirectness, if any exists. Otherwise use only "indirect" or explain what you mean.
Modified as suggested.

L23: which proxy?
We have modified the sentence as follows:
"This allows it to take into account a proxy of the convective cell tops and cloud characteristics." has been modified by:
"As such, it could cover cloud characteristics since the radar was able to measure by means of the principle of electromagnetic wave reflection (Maranan et al., 2018)."

L5: "Note that these wet and dry spells…" this sentence is not necessary here.
This sentence has been removed.

L23: I'd change "extreme long" with "very long". Extreme suggests this is a definition issued from a statistical analysis, which is not the case.

The term "extreme long" is used because these dry spells recorded the extreme values of the PDF (Probability Density Function).

L5: "and because of the synoptic systems associated with the rainfall variability in Senegal" this sentence is not necessary here.

This sentence has been removed.

L10: What do you mean with "defined according to..."?

We have modified the sentence as follows:

"These periods are defined according to the different synoptic components that drive the rainfall variability over Senegal." has been modified by:

"These cumulative wet spells are defined according to different synoptic components such as the 10-20 day variability mode of African monsoon rainfall which stems from coupled regional land-atmosphere interactions (Grodsky and Carton, 2001; Mounier and Janicot, 2004)."

Section 2.1: In this section I have the impression that you use OK and BK as reference datasets, but this is never explicitly said in the manuscript.

We thank the reviewer for this comment. We have clarified through all the manuscript that Block kriging (BK) is used as a reference when assessing datasets.

L18: "seasonal precipitation" which season?

We have modified the sentence as follows:

"The first inter-comparison done between all the datasets focuses on the seasonal cycle of the rainfall." has been modified by:

"The first inter-comparison between all the datasets focuses on total seasonal precipitation from June to October."

L20: why the 0.5 threshold is chosen?

We have decided to add some sentences to clarifie this point. We have modified the sentence as follows:

"To get a fair comparison between the products, all the datasets are regridded into the same grid and only grids where observed values derived from kriging methods are considered significant (i.e., the root mean square of kriging data less than 0.5), are kept (Lloyd and Atkinson, 2001)"
has been modified by:
"The kriging method allows for estimation errors. It takes into account the spatial dependency structure of the data. Based on the kriging error, a critical threshold is established to eliminate pixels when estimated data is not reliable. For this study, the threshold of 0.5 was adopted, based on Lloyd and Atkinson (2001)."

L23-25: It's hard to assess similarities just by visual inspection of Fig. 2, an objective analysis (e.g. computing differences) would be helpful. Why do you say that reanalysis underestimate precipitation? Could it be the other way around, i.e. satellite and in-situ observations overestimating precipitation? What I see is that reanalysis seem to show less precipitation, but this is not underestimation, unless you are comparing with a reference dataset. Please clarify and rephrase.

We have clarified this point by clearly mentioning that BK is called the Reference. We have also computed rmse and correlations to assess similarities and differences between datasets.

L25-28: This statement is unclear. What is the effect of the morphing process on precipitation we see in Fig. 2c? Please clarify.
The sentence has been corrected and rephrased as follows:
"Finally our results for CMORPH confirms the findings of Tian et al. (2007) which showed that the regular smoothing of these precipitation resulting from the "morphing" process could have an effect on the intermittency of precipitation."
has been replaced by:
"The findings show that CMORPH is the product exhibiting the lowest cumulative seasonal rainfall especially in Senegal's southern coastal area compared to other datasets. Indeed in this part of the country, CMORPH records cumulative seasonal rainfall of less than 900 mm, whereas in other datasets rainfall amounts exceed 1,100 mm. This result confirms the
findings of Tian et al. (2007) showing that the regular smoothing of precipitation consequential to the "morphing" process can have an effect on precipitation intermittency (Fig. 2c)."

L1-2: Unclear. Which differences and areas are you discussing here? Where we can see the bias? Please clarify.
We have modified the sentence as follows:
"Because of this importance, a robust network of raingauges (about 24) was used to get a more robust and slightly more complexe structure of the cumulated rainfall from OK and BK. When looking over smaller areas, differences are more important and any of the products is able to get this structure, even if their bias stay low."
has been replaced by:
"Because of this strategic importance, a consequential network of rain gauges (about 24) was used to obtain a more robust estimation of ordinary and block kriging (OK and BK).
Regional-scale rainfall patterns are of particular importance. All products showed a similar magnitude of spatial rainfall variations even though this variation is particularly noticeable across the peanut basin with amounts ranging from 400 to 700 mm."

L7: Why BK should be more adapted to compare with satellite estimations?
We have modified the sentence as follows:
"It is also important to notice the large impact of the kriging techniques with the differences between OK and BK because of their calculations (see equation \ref{e1} and \ref{e2}). Indeed, the ordinary kriging assumes stationarity (i.e. the mean and variance of the values are constant in the spatial field) while block kriging estimates mean values over the block rather than over points \citep{Chen2008Rainfall,Wei2009Rainfall}. Thus, maps of the cumulated rainfall using BK is smoother than OK. It means also that BK should be more adapted to be compared to satellite estimation than OK."
has been replaced by:

" This can be explained by the two kriging methods. BK produces an average rainfall estimate at a given location (considered as a "block"). Whereas OK estimates the rainfall value at a point in a region using data near the estimation location. This means the BK method is akin to satellite measurement techniques which also estimate rainfall on pixels (Fig. 3h and Fig. 4h)."

L19-21: This sentence is too vague. Are you comparing datasets with a reference? Where we can see the overestimation of Heug/small rainfalls?
We have clarified this point by clearly mentioning that BK is called the Reference and this sentence has been corrected and rephrased as follows:
"Due to the absence of observed precipitation, it is difficult to know what is the most accurate products, nevertheless, it seems that products such as CMORPH, during all the dry season, and TAMSAT and NCEP for the dry season from October to December overestimate the "Heug" rainfalls."

have been modified by:

"Given these technical problems, it is even more difficult to declare BK as the most accurate data set. Nevertheless, certain products such as CMORPH, during the entire dry season, and TAMSAT and NCEP during the October to December dry season recorded more "Heug" rainfalls than others datasets (Fig. 5)."

L22-23: what's the reference dataset?
We have clarified this point by clearly mentioning that BK is called the Reference and this sentence has been removed: "TAMSAT generates too much precipitation days in the southern part of the region. This is also visible for the cumulated rainfall."

L24-28: Here you discuss the occurrence of dry days during the rainy season, but there's no way to verify your claims. Are you referring to Fig. 3?
We are sorry for these missleading. We refer to Fig.3 for this discussion and Fig.3 has been added in this paragraph.

Page8
L12-13: In Fig. 4 I see clear differences between BK, TAMSAT and CHIRPS on one the one side and TRMM, CMORPH, CPC, NCEP, ERA5 and OK on the other side for DSl and DSxl. However, clear differences are not evident for DSC10 and DSC20. Any statistical tests are used to assess agreement among datasets?
The sentence has been corrected and rephrased as follows:
"The results highlight a good agreement between TAMSAT and CHIRPS on the one hand and CMORPH and TRMM on the other. This is in agreement with the previous results on the dry days (Fig. 4)." has been replaced by:
"In Fig. 6 clear differences emerge between BK, TAMSAT and CHIRPS on the one hand, where the number of DSl does not exceed 1, and TRMM, CMORPH, CPC, NCEP, ERA5 and OK on the other hand with average per season recordings of 2 DSl. This pattern persists for DSxl, though there are clear differences for DSC10 and DSC20. This fits with the previous findings concerning dry days (Fig. 5)."

L22: OK and BK do actually show large differences for DSl and DSxl. Any comments?
We agree with adding discussion about that. These sentences have been added :
"Finally, BK and OK demonstrate important differences in dry spell occurrences. Indeed, the smoothing effect due to kriging is stronger in OK than BK. This is a direct consequence of the two kriging methods as described above."

L24-25: All the DS* metrics focuses on specific dry spell duration. What do you mean with "DSl is more sensitive during transitional periods"? Please clarify this sentence.
We have modified the sentence as follows:
"Note that, due to their definitions, DSC10, DSC20 and DSxl are very sensitive to the dry season (from November to May), whereas DSl focuses on a specific dry spell duration and is more sensitive during transitional periods (i.e. onset and retreat phase of the rainfall)."
has been replaced by:
"Note that, due to their definitions, DSC10, DSC20 and DSxl are quite sensitive to the dry season (November to May), whereas DSl shows rain breaks between 8 and 14 days. Thus, the end of the breaks is necessarily marked by a rainy day, which would explain their sensitivity during the transition phases (i.e. onset and retreat phase of rainfall) and their misreadings during dry season."

L29: How can you infer a delay of the rainy season from increased occurrence of dry spells in May-July? I'd say that more dry spells increase the probability of late start, but whether this does really happen needs to be demonstrated.

We have removed this sentence :

"A significant result is the delay of the rainy season generated by the two reanalyses."

L29: "ERA5 and NCEP reveal overestimation" in comparison to what?

We have clarified this point by clearly mentioning that BK is called the Reference.

L31-32: "The evolution of DSl etc." this sentence is confusing, please clarify.

We have decided to remove this sentence because we have already clarified this point above.

L17-18: "It is important to notice etc." this sentence is confusing, please clarify.

We have modified the sentence as follows:

"It is important to notice that the differences among the products are comparable to the uncertainties related to the kriging methods of observations."

has been replaced by:

"It is important to note that the differences between OK and BK are linked to uncertainties concerning kriging methods for observations. "

L24-25: I actually see a slightly increasing trend in only ERA5 and CPC. You should assess trend significance statistically.

We did not assess the trend because of the short overlap period (1998-2010).

L32: By stating that TRMM is "better" than other products and closer to observations means that you consider observations as "truth".

We have modified these sentences as follows:

"First TRMM appears to be significantly better, and closer to the observations, than the other products. This is true for all the wet spell definitions. " has been replaced by:

"First TRMM appears to be closer to the OK and BK observations than the other datasets. This is true for all wet spell categories."

L6: "differences are not significant": Did you assess statistical significance?

We have modified this sentence as follows:

"Regarding BK, which is, by definition, associated with smoother datasets, the differences are smaller, but still exist especially for the WSCs." has been replaced by:

"Regarding BK, which is associated with smoother datasets by definition, there are fewer differences, but they do remain, especially for the WSCs."

L3: "Significant differences…": Statistical significance of the results is not actually assessed. "The product resulting from the observations and kriged with the BK method is identified as the reference": This should be motivated and stated at the beginning of the paper.

We have clarified this point by clearly mentioning that BK is called the Reference.

L4: "This is justified by the fact that kriged data are more likely to be compared with satellite observations or model data": This looks as a-posteriori choice. Please clarify.

We have clarified this point by clearly mentioning that BK is called the Reference

L11: "This explanation is therefore probably not sufficient": I don't see any explanation here, you just highlight the outcome of your analysis.
We have modified these sentences as follows:
"For wet periods, it turns out that the products with the lowest intense rainfall are also the finest. This explanation is therefore probably not sufficient. Satellite products combining infrared and microwaves allow good sampling (IR) with better intensity extractions (MO). " has been replaced by:
"For wet spells, it turns out that products with the lowest intense rainfall are also the highest resolution datasets. Therefore, the resolution of datasets is probably an insufficient explanation for these differences. Furthermore, satellite products combining infrared and microwave result in good sampling (IR) with improved intensity extractions (MO)."

L12: "trends" word is misleading, it looks as you refer to the interannual variability.
We have modified these sentences as follows:
"TRMM and CMORPH using this combination show similar trends and are often quite close to in situ observations (OK and BK). " has been replaced by:
"TRMM and CMORPH using this combination show similar skill in detection of wet spell intensity and are often quite close to in situ observations."

L21: what is a "unimodal region"?
We have modified these sentences as follows:
"Indeed, in unimodal regions such as the Sahel, the reanalyses are quite close on the main characteristics of the seasonal rainfall cycle and seasonal evolution. " has been replaced by:
"Indeed, in unimodal regions such as the Sahel, where a unique rainy season is observed from June to October, the reanalyses are quite close to the main characteristics of monthly and annual rainfall. This contrasts with the Gulf of Guinea regions where there is a bimodal rainfall regime. However, reanalyses often show quite significant differences in intra-seasonal rainfall characteristics."

L22: what is the difference between "seasonal cycle" and "seasonal evolution"?
We have modified these sentences as follows:
"Indeed, in unimodal regions such as the Sahel, the reanalyses are quite close on the main characteristics of the seasonal rainfall cycle and seasonal evolution. " has been replaced by:
"Indeed, in unimodal regions such as the Sahel, where a unique rainy season is observed from June to October, the reanalyses are quite close to the main characteristics of monthly and annual rainfall."

Figure 5 and Figure 9: What do the plots actually show? Are data monthly aggregated? How frequencies are computed? Please explain.
According to this comment, we have also
decided to add this sentence in the description of the figures:
"Frequency is defined as a ratio of observed days having recorded dry spells."

Figure 10: Why do some datasets show gaps?
According to this comment, we have also
decided to add this sentence:
"These gaps are explained by the lack of some amounts in datasets."

**Technical corrections**

L2: "from several datasets"

Modified as suggested.

L3: "two are based on reanalysis products"

Modified as suggested.

L4-5: "three are based on raingauge observations: CPC Unified V1.0/RT and a 65 raingauge network that has been reggrided by using two kriging methods, namely Ordinary kriging (OK) and Block kriging (BK)".

Modified as suggested.

L6: delete "same".

Modified as suggested.

L6: "daily cumulated rainfall on a 0.25 degree regular grid"

Modified as suggested.

L21: major floods in 2009 and heavy rains in 2012, references are needed.

Modified as suggested.

L4: Program 2017: This reference is connected with a conflict database, please provide a reference on drought impact.

Modified as suggested.

L5: ARC 2004: Impossible to find the link between funding and 2014 drought in this 53 page document. Please provide a more accurate reference.

Modified as suggested.

L10: "to better understand multi-scale variability of the rainfall regime"

Modified as suggested.

L12: "multi-scale variability"

Modified as suggested.

L15: "illustrating a hydrological cycle intensification"

Modified as suggested.

L28-31: please define TAMSAT, CMORPH algorithm and TRMM 3B42 data.

Modified as suggested.

L5: "This paper aims to provide an inter-comparison between several datasets based on satellite data (TRMM-3B42 V7, TAMSAT V3, CMORPH V1.0, CHIRPS V2.0), reanalysis products (NCEP-CFSR, ERA5), and gauge observations (CPC Unified V1.0/RT, provided by ANACIM the National Agency of Civil Aviation and Meteorology). This Intercomparison focuses on the ability of these datasets to detect dry and wet spells"

Modified as suggested.

L25: T62 resolution is around 1.8 degree, which is much coarser than the 0.31 resolution in Table 1.

We have modified these sentences as follows:
"The NCEP-CFSR is available on the T62 Gaussian grid." has been replaced by:
"The NCEP-CFSR is available on the T382 Gaussian grid."

L26: ERA5 data are provided at 0.25 degree resolution, why I see 0.1 in Table 1?

This is corrected.

L28: 30'000 stations globally? Please define GTS and COOP.
We have modified these sentences as follows:
"The last dataset is fully based on raingauges observation (CPC Unified V1.0/RT). CPC Unified V1.0/RT use gauge reports from over 30,000 stations from multiple sources including GTS, COOP, and other national and international agencies"
have been replaced by:
" The last dataset, CPC Unified V1.0/RT, is fully based on rain gauge observations. It uses the gauge reports of over 30,000 stations worldwide from multiple sources including Global Telecommunication System (GTS), Cooperative Observer Network (COOP) and other national and international agencies"

L32: delete "<".

Modified as suggested.

L19: "used"

Modified as suggested.

L8: "a wet event"

Modified as suggested.

L13: delete Maranan et al. 2018.

Modified as suggested.

L27: "latter"

Modified as suggested.

L13: Fig.3

Modified as suggested.

L15: MW?

Modified as suggested.

L11: provided

Modified as suggested.

L15: please add units to standard deviation and RMSD.

Modified as suggested.

L10: "distribution shows tipping points"

Modified as suggested.

L10-11: "finest resolution"

Modified as suggested.

L22: reanalysis are quite similar?

Modified as suggested.

Figure 2: Please change the colour palette, light yellow may be confounded with white.

Modified as suggested.

Figure 8: please check y-axis label.

"Rainfall accumulated (mm)" has been replaced by:
"Rainfall amount (mm)"

**Author's response to anonymous referee 2**

This manuscript greatly improved after the first round of revision and now illustrates the capability of different rainfall datasets to represent high-impact dry and wet spells rather nicely. Particularly the discussion of good and bad performing datasets in view of their data sources is very informative and this paper importantly illustrates the difficulties in deriving information on drought and rainfall extremes, even from the wealth of rainfall datasets available nowadays. It is therefore an important addition to literature, particularly for West Africa where analyses of extreme weather metrics from available rainfall observations become more and more important but reliability of datasets for such applications is too rarely questioned.

Unfortunately and in spite of the authors stating that the manuscript was properly proof-read and even checked by a native speaker, non-cosmetic corrections are still necessary in high density. I would urge the authors to read the entirety of this manuscript again – to actually do it and to add all the –s and cross the ts. It is rather tedious for a reviewer to do this job and was very much not enjoyable. We all know that if only half of those errors make it through to publication it reduces the trustworthiness and readability (and therefore the impact) of this study, which is completely avoidable.

I have a couple of major comments regarding the dry spell results that need to be clarified and the minor comments need to be addressed before I can recommend this manuscript for publication.

We would like to thank the reviewer for his positive comments and suggestions that will im prove the document. We have modified the document as suggested to give this study a more meaningful impact. Notable improvements have been made and clarifications have been included. Please find the responses to general and specific comments in red.

**Major comments:**

• I have methodological concerns (1) regarding the definition of dry spells causing large spreads at the end of the year, inconsistent with early in the year (Fig5) and (2) whether the inclusion of the dry season into metrics skews the aggregated results for DSC10, DSC20 and DSxl (Fig 4,6,7) away from anything that is relevant regarding "drought hazard" or "high impact event" if they are dominated by the dry season signal. Please clarify the behaviour of (1) and illustrate that the dry season sensitivity is not dominating Fig 4,6,7 results (I.e. I'd like to see Fig4,6, for 'rainy' months only, say April-Nov or similar).

We thank the reviewer for the opportunity to clarify this point in the methodology. Indeed, (1) November and December are the beginning of the dry season, therefore, it is common to record rainfall events especially in the south of the country although they are low rainfall. While, the beginning of the year (January) we are in the height of the dry season. (2) In Figs. 4, 6 and 7 the dry season is not taken into account in our computations.

**General comments:**

• Different from the author's answer, there were in fact no additional maps of any metrics added to the supplementary material that would illustrate the spatial pattern of discussed metrics. Could you please add those as mentioned in the response?

Corrected, the spatial pattern of discussed metrics was added in supplementary material.

• Please add letters to figures and refer to panels in plots consistently throughout the manuscript.

The letters have been added to figures and refer to panels in plots throughout the manuscript.

• Where possible provide some key quantification of discussed discrepancies and biases. Currently the analysis remains predominantly qualitative.

We have decided to add two figures to the main document whereas statistical significance of the results is assessed by the correlations and Root Mean Square Error (RMSE).

**Minor comments (those corrections are unfortunately not exhaustive!):**
P1 line 6 remove first "same"
Modified as suggested.

P1 line 10 "while," -> meanwhile
Modified as suggested.

P1, Line 10: why does this suggest an "early cessation" of precipitation – what would be a normal cessation? Doesn't this rather say that rainfall is more intermittent during the cessation of precipitation ("false onset" expression captures that)
The sentence has been modified as follows.
"All datasets show that dry spells are more frequent at the beginning and end of the rainy season, indicating a false start and early cessation of precipitation."
has been modified by:
"All datasets show that dry spells appear to be more frequent at the start and end of rainy seasons. Thus, dry spell occurrences have a major influence on the duration of the rainy season, in particular through 'false onset' or 'early cessation' of seasons."

P1, LL 10-11: remove "while," , the strongest contrasts between the data products [..] observed _for_ the amplitude
Modified as suggested.

P1, Ll 12-13: remove "quite similar", better to say "show a comparable/similar frequency of wet sequences"
Modified as suggested.

P1, L 18 thE Sahel
Modified as suggested.

P2, l 7: "will increase significantly [..] during the lean season of 2018" what does this mean? Is it projected to increase in the future or did the number increase in 2018?
We have modified the sentence as follows:
"In addition, according to the World Food Program (WFP), Senegal is one of the seven Sahelian countries where the number of food insecure people will increase significantly from the current 314,600 to 548,000 during the lean season of 2018
(WFP, 2018)." has been replaced by:
"In 2018, according to the World Food Programme (WFP), Senegal was one of seven
Sahelian countries with significant increase in numbers of food-insecure people, from 314,600 to 548,000 in the 2018 lean season (WFP, 2018)."

P2 ll14-15: "these hybrid rainy seasons, illustrating a rainfall regime intensification , ARE part of"
Modified as suggested.

P2 l34 remove "although", it's "however"
Modified as suggested.

P3, l7; l28; p4 l3: techniques (same for all other cases of "technic")
Modified as suggested.

P4 l3: remove "is" from "is uses a moving neighborhood"
Modified as suggested.

P4 l5: centered AT the block mean
Modified as suggested.

P4 l7: "has a minimum variance of estimation error" - does that mean it minimises the variance of the estimation error? In any case, please clarify
We have modified the sentence as follows:
"The block value Z is a linear average of the n point estimators and has a minimum variance of estimation error." has been replaced by:
"The Z block value is a linear average of the n point estimators and it has a minimum estimated error variance."

P4 ll 9-10: This should be changed to "we use the kriging RMSE to mask out areas [..]"af etc. The part with "can be possibly masked out" is not fitting since Fig1 indeed shows vast masked out areas, so it is applied rather than "can potentially be applied"
Modified as suggested.

P 4, l32: resampled to 0.25 (remove < ), same p5, line 1
Modified as suggested.

P5 l16: what kind of sensitivity is meant here and how is the analysis of those datasets related to future impacts?
We have clarified this point and we have modified the sentence as follows:
"Moreover, a large number of definitions (depending on the duration of the episodes and their intensities) are used in order to highlight the sensitivity of each datasets and, in the future, their potential impacts."  has been replaced by:
"Moreover, a large number of definitions (related to duration of the episodes and their intensity) are used in order to highlight potential differences between datasets for representing the effect of high-impact events on socio-economic activities.
The following subsections present methods applied to detecting dry/wet spells."

P5, l19 used
Modified as suggested.

P5, l20 precipitation IS lower
Modified as suggested.

P5, l23 descripted -> shown
Modified as suggested.

P5, l23-24 during a specific period(s) - remove s - and IS called
Modified as suggested.

P5 26 when the rainfall IS not sufficient … and therefore DOES not provide..
Modified as suggested.

P5, l28 The results presented in THIS study
Modified as suggested.

P5, l29 from the other durationS
Modified as suggested.

P6, l5 "and because of the synoptic systems associated.." this should be better explained and the characteristics of MCSs (e.g. propagating and therefore extremes rarely stationary etc) mentioned. It's otherwise not clear for people who don't work in the region
According to this comment, we have decided to add these news sentences:
"Indeed, the wet spell duration categories were chosen to correspond to the different synoptic systems causing rain in West Africa. Short wet spells are associated with the so-called '3–5 day' African EasterlyWaves (AEWs). These AEWs are synoptic disturbances known to drive mesoscale convective systems throughout West Africa  Diedhiou et al., 1998;Wu et al., 2013)."

P6, l10-11 "these periods are defined according to the different synoptic components that drive the rainfall variability" again, this is very vague. If there is related reasoning it should be stated explicitly, and those factors at least mentioned (preferably with a reference about the importance of that factor)
We have clarified this point also and we have modified the sentence as follows:
"These periods are defined according to the different synoptic components that drive the rainfall variability over Senegal."  has been replaced by:
"These cumulative wet spells are defined according to different synoptic components such as the 10-20 day variability mode of African monsoon rainfall which stems from coupled regional land-atmosphere interactions (Grodsky and Carton, 2001; Mounier and Janicot, 2004)."

P6. L21: remove "the" from a south-north gradient
Modified as suggested.

P6 l22 in termS of
Modified as suggested.

P6 l22-23 It would be good to state in the text that this is rainfall values for June-October, possibly right in the introductory sentence of the "seasonal rainfall" section
Modified as suggested.

P6 l23 closeD -> remove D
Modified as suggested.

P6 l24 kriged observed precipitation datasets (better: in-situ datasets or rain gauge datasets etc)
Modified as suggested.

P6 l25-26 Our results from CMORPH (..) - there are several language problems in this sentence, please correct (confirmS, "which" showed, "these" precipitation)
What is the result here for CMORPH? I assume this refers to lower seasonal precipitation compared to the gauge-based products but it's not stated. It would be useful to quantify "the results are close" or "underestimating" by giving a percentage range for the rainfall differences between those datasets, or correspondence in pattern correlation or anything that underpins the qualitative statements in this section.
The sentence has been corrected, rephrased and the differences have been quantified.
We have modified the sentence as follows:
"Finally our results for CMORPH confirms the findings of Tian et al. (2007) which showed that the regular smoothing of these precipitation resulting from the "morphing" process could have an effect on the intermittency of precipitation."
has been replaced by:

"The findings show that CMORPH is the product exhibiting the lowest cumulative seasonal rainfall especially in Senegal's southern coastal area compared to other datasets. Indeed in this part of the country, CMORPH records cumulative seasonal rainfall of less than 900 mm, whereas in other datasets rainfall amounts exceed 1,100 mm. This result confirms the
findings of Tian et al. (2007) showing that the regular smoothing of precipitation consequential to the "morphing" process can have an effect on precipitation intermittency (Fig. 2c)."

P6 l32-33 "When looking over smaller areas differences are more important and any of the products is able to get this structure even if their bias stay low" Please correct (language problem) and clarify this sentence, which region this refers to and where biases stay low. I'd assume this means something like "Regional-scale patterns in rainfall are of particular importance. All products seem to approximately agree on the magnitude of spatial rainfall variation. Such variation is particularly pronounced across the peanut basin, for which the bias between rainfall products is low" - again, can "low" be quantified? It seems difficult to assess those statements by just eyeballing the maps and no indication of what the authors refer to.
The sentences have been corrected, rephrased and the differences have been quantified.
We have modified the sentence as follows:
"Because of this importance, a robust network of raingauges (about 24) was used to get a more robust and slightly more complex structure of the cumulative rainfall from OK and BK. When looking over smaller areas, differences are more important and any of the products is able to get this structure, even if their bias stay low."
has been replaced by:
"Because of this strategic importance, a consequential network of raingauges (about 24) was used to obtain a more robust estimation of ordinary and block kriging (OK and BK).
Regional-scale rainfall patterns are of particular importance. All products showed a similar magnitude of spatial rainfall variations even though this variation is particularly noticeable across the peanut basin with amounts ranging from 400 to 700 mm."

P7, l8 closest to BK in intensity - can "closest" be quantified, just to give the reader some idea what the magnitudes here are in terms of biases, agreement etc.
We have added two figures to the main document whereas statistical significance of the results is assessed by the correlations and Root Mean Square Error (RMSE) with BK as reference.

P7, l14 Does this paragraph now refer to Fig3? Reference missing
We are sorry for these missleading. We refer to Fig.3 for this discussion and Fig.3 has been added in this paragraph.

P7, l19 accurate productS – remove S
Modified as suggested.

P7 20-21 On P6, 25-26 it says that CMORPH misses local convective rainfall between scans, resulting in somewhat lower seasonal total rain, and here it says it tends to overestimate small (low-intensity?) precipitation, where the authors say "which would explain why the difference appears here but not when looking at the cumulated rainfall". This can be confusing and would
be worth clarifying. I think this says that high-intensity rain dominates the wet season, of which CMORPH misses events in-between scans, but low-intensity events during the dry season are overestimated. But please state this more clearly.
According to this comment, we have decided to modifie the sentence as follows:
"The reasons could be due to the algorithm of CMORPH, that tends to overestimate the small precipitation (Bruster-Flores et al., 2019), which would explain why the differences appear here but not when looking at the cumulated rainfall (Fig. 1)."
has been replaced by:

"Although high-intensity rain dominates the wet season, CMORPH misses some of these events in-between scans while overestimating low-intensity events in the dry season. One explanation for this could be the CMORPH's algorithm since it tends to be more sensitive to the false alarm rate (FAR), or the fraction not stemming from events detected by the CMORPH algorithm (Bruster-Flores et al.,2019)."

P7 l23: in termS of, better would simply be "This is also visible for the cumulated rainfall"
Modified as suggested.

P7 l27 finalLy
Modified as suggested.

P7 l29: it is a difficult – remove "a", better than "to find the reasons" would be "to suggest an ex-planation for"
Modified as suggested.

P8 l7 different typeS, depending on their
Modified as suggested.

P8 l8-10 "In the main document".. and reference to supplementary can be shortened to "We focus on.." with (see Table 2 for the definitions, further results in supplementary material). "Nevertheless [..]" can be dropped.
Modified as suggested.

P8 l13: dry days is Fig 3, not Fig4
Modified as suggested.

P8 l13 "This is in agreement with the previous result" - as the authors show later on, this is not in agreement regarding the CMORPH / TRMM behaviour, which so not agree for the dry days.
Modified as suggested.

Fig 4 caption: "Boxplots of the average number [..] the left and right edges of the box" this should be bottom and top edges. What does "extreme values" for the whisker position mean. Min and max? Is this really per year or again from June-October like Fig2? If it is per year, wouldn't the dry sea-son performance shown in Fig2 predominantly affect those extreme dry spell indices?
The caption has been clarified and modified as follows:
"Boxplots of average number of dry spells (DSC10, DSC20, DSl and DSxl) per year collected on all grid points for the 9 gridded datasets used ( TRMM, TAMSAT, CMORPH, CHIRPS, CPC, NCEP, ERA5, BK, OK). The - represent the median value, The + represent the mean value, the left and right edges of the box represent the 25th and 75th percentile values, respectively, while the "whiskers" represent the extreme values. The average number of dry spells is computed on the overlap period (1998-2010). Details on the datasets and dry spells are provided in Tables 1 and 2 , respectively."
has been replaced by:
"Box plots of average number of dry spells (DSC10, DSC20, DSl and DSxl) per year collected on all grid points for the 9 gridded datasets used ( TRMM, TAMSAT, CMORPH, CHIRPS, CPC, NCEP, ERA5, BK, OK). The minus sign (-) represents the median value, the plus sign (+) repre-sents the mean value, the bottom and top edges of the box represent the 25th and 75th percentile va-lues, respectively, while the "whiskers" represent the extreme values (5 and 95%). The average number of dry spells is computed on the overlap period (1998-2010). Details on the datasets and dry spells are provided in Tables 1 and 2, respectively."

P8 l16 "than TAMSAT and CHIRPS" replace with "as"
Modified as suggested.

P8 l19 cloud top temperature
Modified as suggested.

P8 l 20 MO was already introduced in l15
Modified as suggested.

P8 ll 21-22 This can explained ...compared to the observations -> This may explain the relative good performance [..] compared to the gauge observations
Modified as suggested.

Fig5: how is this frequency defined? Description in caption and text just says "seasonal cycle of dry spells" without further specification. Also, why are there such inconsistencies moving from Dec to Jan? Particularly visible for DCS10, 20 and DSxl. Is there in problem in how the dry spells are identified at the end of the year? Must be a methodological issue that the spread is large in Dec and gone in Jan. Does this affect the aggregated metrics in the other plots?
According to this comment, the periods of the year which calculation is realized for each plot are now well defined and explicitly mentioned during the description of the Figures. We have also decided to add this sentence to clarify the frequency:
"Frequency is defined as a ratio of observed days having recorded dry spells."

P8 l25: It is a very important point that those dry spell metrics are so strongly affected by the dry season and should be pointed out much earlier in the manuscript. While the behaviour of the datasets during the dry season is interesting (and sufficiently shown in the seasonal cycle plots), the importance of dry spells depends on whether they appear during the wet or dry season. For example, Fig4 shows that DSC20 is around 1 or below per year, questioning the usefulness of this metric in the hazard context. It suggests that this metric reaches "1 occurrence per year",
which likely reflects the dry season - this is not very interesting and not reflecting an "extreme event". On the other hand, it would be an important information if this event occurred once a year during the monsoon season. How much are the dry spell results skewed towards rainfall dataset dry season skill (affected by low-intensity precipitation breaks rather than MCSs)? Why weren't the non-seasonal cycle plots restricted to June-Oct (or at least months outside the dry season)?
We agree with adding discussion about that. Nevertheless, in Figures 4, 6 and 7 we consider only dry spell events that occurred during the rainy season between June and October. These sentences have been modified as follows.
"To better understand these different behaviors, the seasonal evolution is taken into account (Fig. 5) illustrating the feature of the seasonal evolution of the dry spells over Senegal. Note that, due to their definitions, DSC10, DSC20 and DSxl are very sensitive to the dry season (from November to May), whereas DSl focuses on a specific dry spell duration and is more sensitive during transitional periods (i.e. onset and retreat phase of the rainfall)."
have been modified by:
"To better analyze these different behaviors, seasonal progression is taken into account (Fig. 7) illustrating frequency which is defined as a ratio of observed days having recorded dry spells. Note that, due to their definitions, DSC10, DSC20 and DSxl are quite sensitive to the dry season (November to May), whereas DSl shows rain breaks between 8 and 14 days. Thus, the end of the breaks is necessarily marked by a rainy day, which would explain their sensitivity during the transition phases (i.e. onset and retreat phase of rainfall) and their misreadings during dry season."

P8, l31 in agreement FOR the observations
Modified as suggested.

P8 31-34 "The evolution of DSl is also interesting by focusing on relative mild droughts with specific durations that are sensitive to dry spells during the onset and retreat phases of the monsoon. This detection is, by far, the more variable from one product to another. For this specific drought it is difficult to distinguish specific behavior of a group of products. Each possesses a specific time evolution [..]"
Please improve wording. [..] is also interesting as it represents/characterises relatively mild droughts with a fixed duration. This metric is most sensitive to dry spells during [..], and is by far the most variable [..]. For this dry spell metric, it is difficult to distinguish any specific behavior [..]. Each possesses an individual time evolution [..]
Modified as suggested.

P9 ll3-5: gauge observations (the difference to satellite observations is otherwise not clear). Indeed, the difference between the interpolated gauges is remarkable and, if ignoring ERA5, almost as large as the spread between the satellite observations. Again, it would be worth to quantify this uncertainty in the text. Looking at DSl at the hight at the rainy season between Aug-Sep, the frequency difference between BK and OK is around 20%. The dry day frequency increases by more than 100% just changing from BK to OK, based on the same set of stations. Please be more explicit in numbers about statements rather than to rely on handwaving only
Modified as suggested and the term "observations" has been replaced by: "raingauges"

P9 l10: spatial datasets -> gridded datasets
Modified as suggested.

P9 l11: are providing in -> are provided in THE
Modified as suggested.

Fig6 caption: it should be BK which is mentioned as reference dataset here. What is the x-axis? If it is standard deviation too the ticks should be similar to the y-axis.
Modified as suggested.

Is it correct that this diagram was calculated from the spatial maps (like Fig2) of those metrics and e.g. spatially correlated? Which leads me to the question why no metric map was added to the supplementaries (contrary to what was stated in the reviewer response)?
We confirm that Taylor diagrams represent the spatial correlation of dry/wet spells. Spatial maps of metrics have added  in the supplementary material.

Again this relatively good aggregated agreement may be artificially boosted by including the long dry season. What would this look like for the rainy period only (or say April-Nov?). I think it doesn't reflect well what was shown based on the seasonal plots and distracts from the fact that discrepancies are large when it's most important.
In Figs. 4, 6 and 7 we consider only dry spells that occurred during the rainy season between June and October.

P9 l12-13 "For the DSC10 and DSC20 and the DSl there is no clear difference amongst the datasets. However, DSC10 is more sensitive to the datasets."
Is this supposed to refer to DSC20, DSl and DSxl, which all sit in the area of low standard deviation? The spatial correlation for those metrics seems rather low compared to DSC10
This sentence needs to be clarified and we have modified the text as follows:
"For the DSC10 and DSC20 and the DSl there is no clear difference amongst the datasets. However, DSC10 is more sensitive to the datasets."
modified by:

"Spatial correlation is strongest with DSC10, above 0.8 for all datasets, while for the other metrics we find correlations around 0.5 although the dispersion is less marked for DSC20, DSl and DSxl."

P9, l22 similitude -> similarity
Modified as suggested.

P9 ll23-24 "Finally, DSl displays a specific time evolution." -> displays a time evolution that seems distinct from the other metrics?
Modified as suggested.

P9 l32 observations -> in-situ / gauge observations
Modified as suggested.

P10 l3 I would suggest the authors add lettering to their plots and refer to Figx a,b etc throughout the manuscript. That would make it much easier to follow which panel is being discussed without having to check and recheck the acronyms.
Modified as suggested.

P10 l7 I think that should read "WS1 99P" instead of WSl
Modified as suggested.

P10 l10 "This distribution shows to see tipping points on daily rainfall." language problem, please rephrase
We have modified the sentence as follows:
"This distribution shows to see tipping points on daily rainfall."
has been replaced by:
"This figure illustrates relatively well how the intensity of daily rainfall can be detected via datasets."

P10, l21 except to the -> except FOR the WSM
Modified as suggested.

P10 ll24-25 contributes bias correction -> allows for such biases to be taken into account
Modified as suggested.

P11 l3 in-situ observations
Modified as suggested.

P11 l4-5 are more likely to be compared with -> are more likely to be comparable to gridded [..]
Modified as suggested.

P11 l25 the monitoring [..] are compared -> the monitoring [..] is tested OR the representation [..] is compared
Modified as suggested.

P11 l26 3 products BASED on raingauges
Modified as suggested.

P11 l27 by upgrading or -> by area averaging, interpolation or[..]
Modified as suggested.

P11 l29 THE large-scale climatology
Modified as suggested.

P11 l33 for an average rainfall like most of -> remove "an", like FOR most of
Modified as suggested.

L33 this good agreement start to dissipate -> startS
Modified as suggested.

P12, l2 "It turned out that each of the kriging methods were positioned in these groups." -> Interestingly, from the kriging methods each falls into one of these groups.
Modified as suggested.

P12 l10 "However, there is less agreement between the different data products for dry spells than for the wet spells." Shouldn't this be "there is MORE agreement for dry spells than for wet spells" ?
Modified as suggested.

---

## Author Response (AR3)

I think this manuscript on wet and dry spells in Senegal as derived from different datasets is now in an acceptable state and a useful contribution to West African literature. I have a few last minor comments that I hope will be addressed before publication:

We thank again the reviewer for providing these useful comments which allow the paper to reach the standard of the journal. We took into account all of them to substantially improve the quality of this study. We have replied to all his/her minors comments in red.

• Introduction, p3 ll6-11: "The reference dataset chosen […] We therefore suggest BK as the best available reference candidate" This new added paragraph seems misplaced in the introduction and should be moved into the "Rain Gauge Data and Kriging Methods" section to after the kriging methods have been explained. The justification for the reference dataset seems much more fitting here.

According to this comment, we have decided to move the new sentence from the Introduction to the "Rain Gauge Data and Kriging Methods" section to after the kriging methods have been explained.

• Dataset description: Reading through the satellite dataset description again, it would be better to be more precise about the input that is used to create different datasets, for example:
p4 ll 17-18: "TRMM-3B42 V7 and CMORPH V1.0 are exclusively satellite based" - this is not entirely true. TRMM-3B42 V7 uses GPCC monthly gauge rainfall for a bias adjustment of derived MW / IR rainfall estimates. Similarly, only CMORPH-RAW does not use gauge information (it's not further specified in the manuscript which CMORPH product is used). I suggest not using "exclusively" when it's not in fact exclusive.

Thank you for the clarification. Modified as suggested

ll19-21: " TRMM-3B42 V7 and CMORPH V1.0 are characterized by combining infrared and microwave measurements while CHIRPS and TAMSAT exclusively use infrared measurement techniques."

Again, the wording is not accurate. CHIRPS uses TRMM3B42 to calibrate the rainfall estimation from cold cloud duration and uses gauge data for bias adjustment, while TAMSAT exploits gauges to generate their climatological calibration. Wording like "primarily based on thermal-infrared" while acknowledging other data sources would be more appropriate.
A useful recent overview is given in Le Coz and Van De Giesen (2020), Journal of Hydrometeorology

We thank the reviewer for these comments. We have modified the text as follows:

"TRMM-3B42 V7 and CMORPH V1.0 are exclusively satellite based, while CHIRPS V2.0 and TAMSAT V3 combine both raingauges and reanalyses with satellite data (Kummerow et al., 1998; Nesbitt et al., 2006; Huffman et al., 2007). TRMM-3B42 V7 and CMORPH V1.0 are characterized by combining infrared and microwave measurements while

CHIRPS and TAMSAT exclusively use infrared measurement techniques (Funk et al., 2015; Maidment et al., 2017)."

have been replaced by:

"TRMM-3B42 V7 and CMORPH V1.0 are characterized by combining infrared and microwave measurements while CHIRPS and TAMSAT primarily based on thermal-infrared measurement techniques (Kummerow et al., 1998; Nesbitt et al., 2006; Huffman et al., 2007, Funk et al., 2015; Maidment et al., 2017). Recently, Le Coz and Van De Giesen (2020) provide a detailed overview of these products and their recommendations to detect different types of hazards."

• In my previous review, I pointed out that in Figure 7 a,b,d the dry spell frequencies in Nov/Dec do not seem realistic when compared to Jan-March. If this was plotted continuously through the dry season (rather than stopping in Dec) the Dec-Jan transition would show a sudden step change. The authors explained the difference with "there being some rain in Nov/Dec while Jan is the peak of the dry season", which I think isn't a valid answer here. It does not explain the Dec/Jan step change, which reaches almost 40% for (d) Dsxl for ERA5 and just does not seem realistic (Fig5 does not suggest such jumps). However, since the dry season does not affect the main results I would just ask the authors again to check the "end of year"-handling of their code and if needed to correct Fig7 accordingly.

We thank the reviewer for pointing out that problem. Indeed, after careful verification and as suspected, we have detected an effect due to the end of the year and the too large window of smoothing. We have corrected that problem and we provide a new graph with correct values and a smaller smoothing window.

• There are still frequent, mostly minor, language issues, which I won't comment on any more – I don't know whether some of this can still be resolved at the editorial level. However, I definitely suggest using "similarity" or "correspondence" for all cases of "similitude"/ "affinity" in the conclusion.

Again, we are sorry about that. The document has been corrected by a native english speaker (we still have the receipt). We have also corrected the suggested modifications.